# Evolving Standardization for Continual Domain Generalization over Temporal Drift

**Mixue Xie**[1]    **Shuang Li**[1,*]    **Longhui Yuan**[1]    **Chi Harold Liu**[1]    **Zehui Dai**[2]

[1]Beijing Institute of Technology, China    [2]Lazada Search & Monetisation Tech, China

{mxxie,shuangli,longhuiyuan}@bit.edu.cn, liuchi02@gmail.com

zehui.dzh@alibaba-inc.com

## Abstract

The capability of generalizing to out-of-distribution data is crucial for the deployment of machine learning models in the real world. Existing domain generalization (DG) mainly embarks on offline and discrete scenarios, where multiple source domains are simultaneously accessible and the distribution shift among domains is abrupt and violent. Nevertheless, such setting may not be universally applicable to all real-world applications, as there are cases where the data distribution gradually changes over time due to various factors, e.g., the process of aging. Additionally, as the domain constantly evolves, new domains will continually emerge. Re-training and updating models with both new and previous domains using existing DG methods can be resource-intensive and inefficient. Therefore, in this paper, we present a problem formulation for *Continual Domain Generalization over Temporal Drift* (CDGTD). CDGTD addresses the challenge of gradually shifting data distributions over time, where domains arrive sequentially and models can only access the data of the current domain. The goal is to generalize to unseen domains that are not too far into the future. To this end, we propose an *Evolving Standardization* (EvoS) method, which characterizes the evolving pattern of feature distribution and mitigates the distribution shift by standardizing features with generated statistics of corresponding domain. Specifically, inspired by the powerful ability of transformers to model sequence relations, we design a multi-scale attention module (MSAM) to learn the evolving pattern under sliding time windows of different lengths. MSAM can generate statistics of current domain based on the statistics of previous domains and the learned evolving pattern. Experiments on multiple real-world datasets including images and texts validate the efficacy of our EvoS.

## 1  Introduction

In real-world applications, the assumption that training and testing data conform to the same distribution, a prerequisite for the success of contemporary deep learning methods, is seldom valid. A prime example of this can be found in autonomous driving, where a vehicle may traverse environments that switch from daylight to nightfall or from urban to rural. As environmental conditions change, the issue of distribution shift [5, 4, 51, 62] arises. Moreover, directly utilizing a model trained on in-distribution (ID) data in an out-of-distribution (OOD) context often results in catastrophic performance deterioration. Consequently, ensuring that models perform well on OOD data has emerged as a crucial challenge for the widespread deployment of machine learning models in the real world.

To cope with the distribution shift, two dominant paradigms have been systematically explored depending on the availability of target (test) domain. One is domain adaptation (DA), which aims to assist the model learning on an unlabeled or label-scarce target domain by transferring the knowledge

---

*Corresponding author.

from a related and label-rich source domain [15, 34, 29, 35]. Yet, target data are not always known or available in advance. The other is domain generalization (DG), the goal of which is to learn a model capable of generalizing to any unseen domain by using date from multiple related but distinct source domains [62, 36, 27, 63, 26]. Though the second paradigm DG has attained some encouraging outcomes, its configuration is limited to offline and discrete scenarios where multiple source domains can be accessed simultaneously, and the distribution shift among domains is sudden and severe. For instance, the prevalent benchmark PACS [25] in current DG methods comprises four distinct domain styles - "Art", "Cartoon", "Photo" and "Sketch". Nevertheless, this type of configuration may not be suitable for all real-world applications. There are also some cases that the data distribution gradually evolves over time, the distribution shift arising from which is referred to as *temporal drift* [56].

On one hand, the real-world scenarios often exhibit underlying evolutionary patterns [59, 44]. For example, annual or monthly weather data can be utilized for weather forecasting [12]. The evolutionary patterns can be exploited to enhance generalization capabilities towards future unseen domains that are not too distant. However, current DG methods often fail to consider these patterns, leading to suboptimal performance. On the other hand, since data distribution is constantly evolving, new domains will continue to emerge. Consequently, it is imperative to efficiently utilize these new domains to further enhance the model's performance for practical applications. For instance, in the context of advertisement recommendation, user browsing data for various products continually surfaces. How can we leverage these newly collected data to enable more accurate advertisement recommendations tailored to each user's preferences in the days to come? One simple approach is to store data from previous domains and retrain the model using both the new and previous domains with the existing DG techniques. However, such way may consume huge training resources due to the accumulation of data and is of low efficiency, especially for scenarios with rapid data accumulation.

In this paper, we formulate the aforementioned problems as Continual Domain Generalization over Temporal Drifts (CDGTD), where the data distribution gradually drifts with the passage of time and the domain arrives sequentially. And the goal of CDGTD is to generalize well on future unseen domains that are not too far under the circumstance that only current domain is available at current times point while data from previous domains are inaccessible. Although some temporal / evolving DG methods [44, 59, 3, 41] have been proposed to handle the temporal drift, most of them [59, 44, 41] works in non-incremental setting, i.e., multiple domains can be accessed simultaneously. Instead, we design an Evolving Standardization (EvoS) approach specialized for CDGTD.

For CDGTD, there are two main challenges: how to characterize the evolving pattern in the incremental setting and how to achieve generalization using the learned evolving pattern. For the former, we draw inspiration from the sequence modeling capability of transformers [31] and design a multi-scale attention module (MSAM) to learn the evolving pattern underlying the feature distribution of domains. Specifically, we store the statistics (i.e., mean and variance of features) of previous domains and use sliding time windows of different lengths over the statistic tokens to obtain multi-scale information. Then multi-scale statistic tokens are fed into MSAM to generate statistics of current domain. MSAM is trained over the whole sequence of domains to learn the evolving pattern. Here, we integrate multi-scale information, with the consideration that some evolving patterns may be better characterized at different time intervals, e.g., seasonal climate and daily weather. For the latter challenge, in order to mitigate the temporal drift, each domain is transformed into a common normal distribution by conducting feature standardization with the generated statistics of corresponding domain. Besides, considering that the feature encoder may suffer from catastrophic forgetting and overfitting to current domain, we constrain it to learn a shared feature space among domains via the adversarial learning.

**Contributions: 1)** We formulate a promising but challenging problem of continual domain generalization over temporal drift (CDGTD), which has seldom been explored, compared with traditional DG. **2)** An evolving standardization (EvoS) approach is specially proposed for CDGTD, which can characterize the evolving pattern and further achieve generalization by conducting the feature standardization. **3)** Experiments on multiple real-world datasets with different models verify the effectiveness of EvoS and the flexibility to be applied on different models.

## 2   Related Work

**Domain Generalization (DG)** aims to learn a model that can generalize well to any unseen target domains by leveraging data from multiple source domains. In recent years, a wide range of DG

methods have been proposed [6, 39, 27, 26, 63, 19, 37]. According to the strategies of improving generalization, existing DG methods roughly fall into the three categories. Representation learning based methods [7, 39, 50, 27, 27, 2, 17, 42] aim to learn domain-invariant or domain-shared representations to enable models to generalize to unseen target domains. Data augmentation/generation based methods [53, 61, 63, 57, 32] focus on manipulating inputs to facilitate learning general representations. Differently, other DG methods instead employ general learning strategies like meta-learning [26] and ensemble learning [21] to improve generalization ability. With full access to all source domains, these DG methods have achieved promising performance on unseen domains with abrupt and violent distribution shifts. Unfortunately, without the consideration of the evolutionary pattern of domains over time, existing DG methods are usually less efficient under the scenario of temporal drift.

**Continual Learning (CL)** is a machine learning paradigm, where models learn continuously from a stream of tasks over time and meanwhile try to retain performances on all seen tasks [22, 1, 24, 9, 46, 47]. Existing CL methods can be roughly categorized into replay-based [46, 10, 52], regularization-based [22, 60, 30] and structure-based [45, 14] methods. In addition, the combination of CL and DA has attracted lots of attention [23, 17, 33, 43, 55]. Yet, combining CL and DG remains underdeveloped. In this work, we propose CDGTD which continually trains the model on sequential domains, but the goal is to generalize well on novel domains in the near future. This distinct objective from CL yields different challenges: the modeling of underlying evolutionary patterns of temporal domains and how to utilize these patterns to mitigate the distribution shifts in forthcoming domains.

**Test Time Adaptation (TTA)** focuses on adapting the source-pretrained model during testing phase with test data [40, 49, 20, 54, 58]. For example, [49] corrects the activation statistics of batch normalization using test data. T3A [20] uses a back-propagation-free manner to adjust only the weights of the linear classifier during test time. By contrast, CDGTD optimizes the model during training phase with sequential domains and emphasizes on generalization without using test data.

**Evolving / Temporal Domain Generalization** has emerged in recent years, aiming to tackle the problem of generalizing on temporally drifted domains, where the environment evolves dynamically over time [41, 3, 44, 59]. GI [41] proposes a time-sensitive model architecture to capture the time-varying data distribution and the model is supervised with the first-order Taylor expansion of the learned function to advance the generalization in the near feature. DRAIN [3] launches a Bayesian framework to model the concept drift and utilizes a recurrent neural network to dynamically generate network parameters to adapt the evolving pattern of domains. LSSAE [44] incorporates variational inference to explore the evolving patterns of covariate shift and label shift in the latent space. The goal of learning evolutionary patterns from source domains and generalizing to domains in the near future is similar to our EvoS. The main difference is that the model in our EvoS is incrementally trained on sequentially arriving domains, considering the low efficiency of "offline" training with accumulated domains. By contrast, the aforementioned methods [41, 44] require multiple source domains to be simultaneously available. Besides, the Taylor expansion in [41], variational inference in [44] and network parameters generation in [3] make them hard to expand on large models.

## 3 Evolving Standardization for Continual Domain Generalization over Temporal Drift

### 3.1 Problem Formulation of CDGTD

Here, we take the $C$-class classification problem as an example, where $X$ and $Y$ denote the data and label space, respectively. Suppose that $T$ source (training) domains $\{\mathcal{D}^1, \mathcal{D}^2, \cdots, \mathcal{D}^T\}$ sequentially arrive, which are sampled from distributions at $T$ different times points $\mathsf{t}_1 < \mathsf{t}_2 < \cdots < \mathsf{t}_T$. At time point $\mathsf{t}_t$, $t \in \{1, 2, \cdots, T\}$, only the domain $\mathcal{D}^t = \{\boldsymbol{x}_i^t, y_i^t\}_{i=1}^{N^t}$ is accessible, where $\boldsymbol{x}_i^t \in X$, $y_i^t \in Y$ and $N^t$ is the number of training samples in $\mathcal{D}^t$. Previous and future domains are unavailable. In addition, as previous temporal/evolving DG methods [44, 41, 3], we further assume that the data distribution $P(X, Y)$ of domains evolves temporally, i.e., the distribution of domains changes along the time following certain patterns. The goal of CDGTD is to enable the model, composed of a feature encoder $\mathcal{E} : \boldsymbol{x} \rightarrow \boldsymbol{f} \in \mathbb{R}^{d_f}$ ($d_f$ is the dimension of features) and a classifier $\mathcal{C} : \boldsymbol{f} \rightarrow y \in \{0, 1, \cdots, C - 1\}$, to generalize well on $K$ unseen target (test) domains in the near future: $\{\mathcal{D}^{T+k}\}_{k=1}^K$. To this end, two main challenges need to be addressed. One is to characterize the evolving pattern of domains, which we address by designing a multi-scale attention module (MSAM)

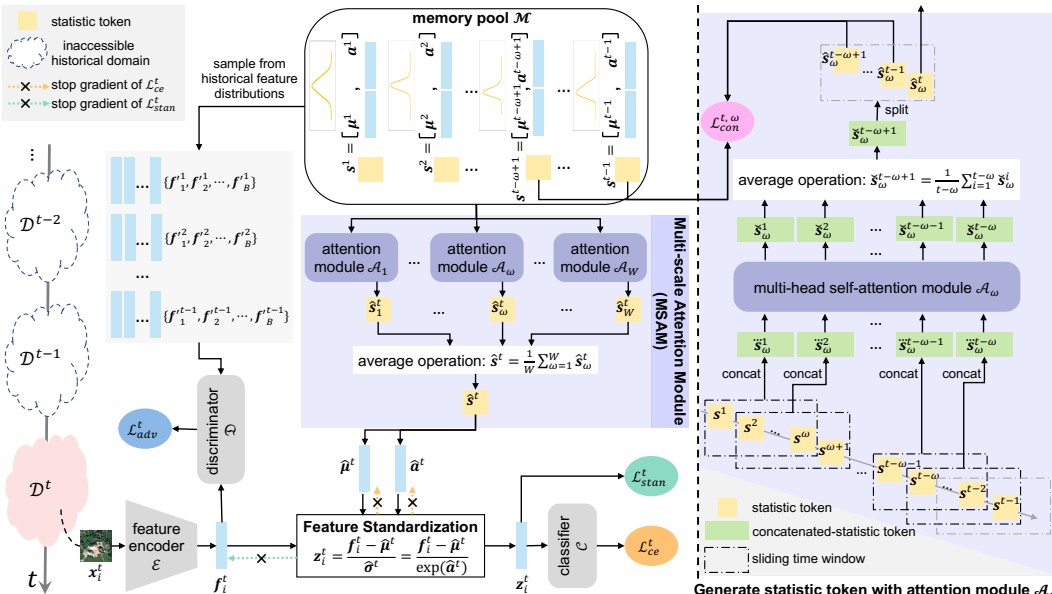

Figure 1: Overview of EvoS. Memory pool $\mathcal{M}$ stores previously generated domain statistics (i.e., $\boldsymbol{\mu}^i$ (mean) and $\boldsymbol{a}^i$ (logarithm of standard deviation) of features, $1 \le i \le t-1$). With the $t$-th domain $\mathcal{D}^t$ available, loss $\mathcal{L}_{stan}^t$ is minimized to train the multi-scale attention module (MSAM) to generate statistics $\hat{\boldsymbol{\mu}}^t$ and $\hat{\boldsymbol{a}}^t$ that approach the real ones via leveraging historical statistics in $\mathcal{M}$. Besides, we sample features from historical feature distributions as the proxy of previous domains to conduct adversarial training with current domain, encouraging to learn a shared feature space.

inspired from the sequence modeling ability of transformers. The other is the generalization, which is realized by transforming domains into a common normal distribution via the feature standardization.

## 3.2 EvoS: Evolving Standardization

**Single-scale Attention.** Inspired by the capability of the transformer to model the relationships among sequences [31], we introduce its attention mechanism to model the evolving pattern among sequential domains. Before presenting our multi-scale attention module (MSAM), we first introduce how single-scale attention works. Without loss of much generality, we assume that the feature distribution of domain $\mathcal{D}^t$ follows the normal distribution, which is characterized by the mean vector $\boldsymbol{\mu}^t \in \mathbb{R}^{d_f}$ and the standard deviation vector $\boldsymbol{\sigma}^t \in \mathbb{R}^{d_f}$. Now, our goal turns out to be learning the evolving pattern of the feature statistics (i.e., $\boldsymbol{\mu}^t$ and $\boldsymbol{\sigma}^t$). In practice, to ensure that $\boldsymbol{\sigma}^t$ is non-negative, we choose to learn $\boldsymbol{a}^t$ (the logarithm of $\boldsymbol{\sigma}^t$). In this way, $\boldsymbol{\sigma}^t = \exp(\boldsymbol{a}^t)$ is always non-negative. Besides, considering the unavailability of previous domains, we store the learned statistics at time point $t_t$ into a memory pool $\mathcal{M}$ for future uses once the training procedure finishes on domain $\mathcal{D}^t$. Having the feature statistics of previous $t-1$ domains ready, we then utilize the multi-head self-attention module $\mathcal{A}$ to generate the statistics of domain $\mathcal{D}^t$ at time point $t_t$.

Concretely, given statistic tokens $\{\boldsymbol{s}^i\}_{i=1}^{t-1}$ of previous $t-1$ domains, where $\boldsymbol{s}^i = [\boldsymbol{\mu}^i, \boldsymbol{a}^i] \in \mathbb{R}^{2d_f}$ is the token of concatenated statistics from domain $\mathcal{D}^i$, the output of $\mathcal{A}$ is expressed as

$$\hat{\boldsymbol{S}}^t = \mathcal{A}(\boldsymbol{S}^{t-1}) = [\mathrm{SA}_1(\boldsymbol{S}^{t-1}), \mathrm{SA}_2(\boldsymbol{S}^{t-1}), \cdots, \mathrm{SA}_{n_h}(\boldsymbol{S}^{t-1})]\mathbf{W}_{fc}, \quad \mathbf{W}_{fc} \in \mathbb{R}^{(n_h \cdot d_h) \times 2d_f},$$
$$= [\hat{\boldsymbol{s}}^1; \hat{\boldsymbol{s}}^2; \cdots; \hat{\boldsymbol{s}}^{t-1}],$$
$$\boldsymbol{S}^{t-1} = [\boldsymbol{s}^1; \boldsymbol{s}^2; \cdots; \boldsymbol{s}^{t-1}], \quad \boldsymbol{S}^{t-1} \in \mathbb{R}^{(t-1) \times 2d_f} \tag{1}$$

where $\hat{\boldsymbol{s}}^i$ is the $i$-th output statistic token of $\mathcal{A}$, $n_h$ and $d_h$ are the number and feature dimension of heads in the multi-head self-attention module and $\mathbf{W}_{fc}$ is the learnable parameters of $\mathcal{A}$ to convert feature dimensions. $\mathrm{SA}_i(\cdot)$ is the self-attention of the $i$-th head, which operates as follows:

$$[\boldsymbol{S}_{i,q}^{t-1}, \boldsymbol{S}_{i,k}^{t-1}, \boldsymbol{S}_{i,v}^{t-1}] = \boldsymbol{S}^{t-1}\mathbf{W}_{qkv}^i, \quad \mathbf{W}_{qkv}^i \in \mathbb{R}^{2d_f \times 3d_h}$$
$$\mathrm{SA}_i(\boldsymbol{S}^{t-1}) = \mathrm{softmax}(\boldsymbol{S}_{i,q}^{t-1}\boldsymbol{S}_{i,k}^{t-1}{}^\top/\sqrt{d_h})\boldsymbol{S}_{i,v}^{t-1}, \tag{2}$$

where $\mathbf{W}_{qkv}^i$ is the learnable parameters of the $i$-th head and $\boldsymbol{S}_{i,q}^{t-1}, \boldsymbol{S}_{i,k}^{t-1}, \boldsymbol{S}_{i,v}^{t-1}$ are the query, key and value embeddings of $\boldsymbol{S}^{t-1}$. Finally, we use the average of all output statistics tokens of the attention module $\mathcal{A}$ as the generated statistic token $\hat{\boldsymbol{s}}^t$ for time points $t_t$:

$$[\hat{\boldsymbol{\mu}}^t, \hat{\boldsymbol{a}}^t] = \hat{\boldsymbol{s}}^t = avg(\hat{\boldsymbol{S}}^t) = \frac{1}{t-1} \sum_{i=1}^{t-1} \hat{\boldsymbol{s}}^i. \tag{3}$$

By continuously generating new statistics from historical ones throughout the whole sequence of domains, the attention module $\mathcal{A}$ is expected to learn the evolving pattern of feature distributions. Note that we directly introduce learnable statistic vectors $\hat{\boldsymbol{\mu}}^1, \hat{\boldsymbol{a}}^1$ and $\hat{\boldsymbol{\mu}}^2, \hat{\boldsymbol{a}}^2$ for time points $t_1$ and $t_2$, respectively, since the tokens of historical statistics are not enough for the attention module $\mathcal{A}$ to work. That is, the attention module $\mathcal{A}$ is only used at time points $t_t, t \geq 3$.

**Multi-scale Attention Module (MSAM).** The above details how to model the evolving pattern using single-scale attention. However, it is limited to learn the pattern using a sliding time window of length 1 over the statistic tokens. Sometimes, the evolutionary pattern may be better captured by considering longer time intervals. For instance, the data of a season in each time window would be more suitable for capturing the evolving pattern of seasonal climate, rather than data of a single day in each time window. Considering this, we introduce multi-scale attention in Fig. 1, which leverages information from observation windows of different lengths to better model the evolving pattern.

Specifically, for the multi-head self-attention module $\mathcal{A}_w$ responsible for the time window of length $w$, we slide the time window with stride 1 over historical $t-1$ statistics tokens and concatenate the statistic tokens in each window to obtain the input $\ddot{\boldsymbol{S}}_w^{t-1}$ for the attention module $\mathcal{A}_w$ at time point $t_t$:

$$\ddot{\boldsymbol{s}}_w^i = [\boldsymbol{s}^i, \boldsymbol{s}^{i+1} \cdots, \boldsymbol{s}^{i+w-1}], \quad \ddot{\boldsymbol{s}}_w^i \in \mathbb{R}^{w \cdot 2d_f}, \quad (1 \leq i \leq t-w) \wedge (t \geq w+2)$$
$$\ddot{\boldsymbol{S}}_w^{t-1} = [\ddot{\boldsymbol{s}}_w^1; \ddot{\boldsymbol{s}}_w^2; \cdots; \ddot{\boldsymbol{s}}_w^{t-w}], \quad \ddot{\boldsymbol{S}}_w^{t-1} \in \mathbb{R}^{(t-w) \times (w \cdot 2d_f)}. \tag{4}$$

And similarly, we use the average output $\breve{\boldsymbol{s}}_w^{t-w+1}$ of the attention module $\mathcal{A}_w$ as the generated statistics for the sliding time window at the next time point, the formulation of which is expressed as

$$\breve{\boldsymbol{S}}_w^t = \mathcal{A}_w(\ddot{\boldsymbol{S}}_w^{t-1}) = [\breve{\boldsymbol{s}}_w^1; \breve{\boldsymbol{s}}_w^2; \cdots; \breve{\boldsymbol{s}}_w^{t-w}], \quad \breve{\boldsymbol{S}}_w^t \in \mathbb{R}^{(t-w) \times (w \cdot 2d_f)}$$
$$\breve{\boldsymbol{s}}_w^{t-w+1} = avg(\breve{\boldsymbol{S}}_w^t) = \frac{1}{t-w} \sum_{i=1}^{t-w} \breve{\boldsymbol{s}}_w^i, \quad \breve{\boldsymbol{s}}_w^{t-w+1} \in \mathbb{R}^{w \cdot 2d_f}. \tag{5}$$

Note that in MSAM, the attention module $\mathcal{A}_w$ is to predict the statistics in the next sliding time window of length $w$, instead of one statistic token as in the single-scale attention. This may encourage the attention module to also capture the domain relationships within the window.

Then, we split $\breve{\boldsymbol{s}}_w^{t-w+1}$ into $w$ parts: $[\hat{\boldsymbol{s}}_w^{t-w+1}, \cdots, \hat{\boldsymbol{s}}_w^{t-1}, \hat{\boldsymbol{s}}_w^t] = \breve{\boldsymbol{s}}_w^{t-w+1}$, where $\hat{\boldsymbol{s}}_w^j \in \mathbb{R}^{2d_f}$ can be regarded as the predicted statistic token for time point $t_j$ using the attention module $\mathcal{A}_w, j = t-w+1, \cdots, t-1, t$. Finally, the generated statistic token $\hat{\boldsymbol{s}}^t$ for time point $t_t$ using MSAM is denoted as the average of predicted statistic tokens for time point $t_t$ at different time window lengths:

$$[\hat{\boldsymbol{\mu}}^t, \hat{\boldsymbol{a}}^t] = \hat{\boldsymbol{s}}^t = \frac{1}{W} \sum_{w=1}^{W} \hat{\boldsymbol{s}}_w^t, \tag{6}$$

where $W$ is the maximum length of the sliding time window. Generally, MSAM integrates evolving patterns learned at different scales, contributing better estimation of future feature distributions.

**Feature Standardization.** For CDGTD, the second main challenge is generalization. To this end, we leverage the generated feature statistics by MSAM to transform the distribution of corresponding domain into a standard normal distribution, by which the temporal drift is mitigated. Concretely, for feature $\boldsymbol{f}_i^t = \mathcal{E}(\boldsymbol{x}_i^t)$, its standardized feature $\boldsymbol{z}_i^t$ via the feature standardization is formulated as

$$\boldsymbol{z}_i^t = \frac{\boldsymbol{f}_i^t - \hat{\boldsymbol{\mu}}^t}{\hat{\boldsymbol{\sigma}}^t} = \frac{\boldsymbol{f}_i^t - \hat{\boldsymbol{\mu}}^t}{\exp(\hat{\boldsymbol{a}}^t)}. \tag{7}$$

Another benefit of feature standardization is that it allows classifier to be trained on a common feature distribution, thus avoiding the problem of overfitting to current domain and catastrophic forgetting.

### 3.3 Model Training

This section outlines the four losses $\mathcal{L}_{ce}^t$, $\mathcal{L}_{stan}^t$, $\mathcal{L}_{con}^t$ and $\mathcal{L}_{adv}^t$ involved in the training stage at time point $t_t$. The first one is the essential loss to supervise the model learning for specific tasks, e.g., the following cross-entropy loss $\mathcal{L}_{ce}^t$ for classification tasks in this paper.

$$\min_{\mathcal{E},\mathcal{C}} \mathcal{L}_{ce}^t = \frac{1}{N^t} \sum_{i=1}^{N^t} \mathcal{CE}(\boldsymbol{p}_i^t, y_i^t), \quad \boldsymbol{p}_i^t = \text{softmax}\left(\mathcal{C}\left(\frac{\boldsymbol{f}_i^t - \text{sg}(\hat{\boldsymbol{\mu}}^t)}{\exp(\text{sg}(\hat{\boldsymbol{a}}^t))}\right)\right), \tag{8}$$

where $\mathcal{CE}(\cdot,\cdot)$ is the cross-entropy. Here, stopping gradient $\text{sg}(\cdot)$ is adopted to stabilize training.

Losses $\mathcal{L}_{stan}^t$ and $\mathcal{L}_{con}^{t,w}$ are responsible for the training of the attention module $\mathcal{A}_w$ at time point $t_t$, $w = 1, 2, \cdots, W$. The former loss $\mathcal{L}_{stan}^t$ in Eq. 10 is minimized to ensure that the standardized feature $\boldsymbol{z}_i^t$ follows a standard normal distribution. Specifically, we first calculate the mean vector and variance vector of the standardized features as

$$mean : \hat{\boldsymbol{\mu}}^t = \frac{1}{N^t} \sum_{i=1}^{N_t} \frac{\text{sg}(\boldsymbol{f}_i^t) - \hat{\boldsymbol{\mu}}^t}{\exp(\hat{\boldsymbol{a}}^t)}, \quad variance : \hat{\boldsymbol{v}}^t = \frac{1}{N^t - 1} \sum_{i=1}^{N_t} \left(\frac{\text{sg}(\boldsymbol{f}_i^t) - \hat{\boldsymbol{\mu}}^t}{\exp(\hat{\boldsymbol{a}}^t)} - \hat{\boldsymbol{\mu}}^t\right)^2. \tag{9}$$

Then the loss $\mathcal{L}_{stan}^t$ at time point $t_t$ is expressed as

$$\begin{cases} \min\limits_{\hat{\boldsymbol{\mu}}^t, \hat{\boldsymbol{a}}^t} \mathcal{L}_{stan}^t = \|\hat{\boldsymbol{\mu}}^t - \mathbf{0}\|_2 + \|\hat{\boldsymbol{v}}^t - \mathbf{1}\|_2, & 1 \leq t \leq 2 \\ \min\limits_{\mathcal{A}_w} \mathcal{L}_{stan}^t = \|\hat{\boldsymbol{\mu}}^t - \mathbf{0}\|_2 + \|\hat{\boldsymbol{v}}^t - \mathbf{1}\|_2, & t \geq w + 2 \end{cases}, \tag{10}$$

where $d_f$ is the dimension of features and $w$ is the length of the sliding time window. Here, we just want the learnable/generated $\hat{\boldsymbol{\mu}}^t, \hat{\boldsymbol{a}}^t$ to approach the statistics of true feature distribution, so stopping gradient operation $\text{sg}(\cdot)$ is conducted on $\boldsymbol{f}_i^t$ to prevent the gradient of $\mathcal{L}_{stan}^t$ flowing to the feature encoder $\mathcal{E}$. Otherwise, the gradient of $\mathcal{L}_{stan}^t$ may distort the feature distribution. Ideally, if $\mathcal{L}_{stan}^t$ is minimized, the generated statistics $\hat{\boldsymbol{\mu}}^t$ and $\exp(\hat{\boldsymbol{a}}^t)$ by our MSAM would be equal to the mean vector and standard deviation vector of the true feature distribution of $\mathcal{D}^t$. This means that our MSAM can predict future feature distributions based on historical feature statistics.

And the loss $\mathcal{L}_{con}^t$ is to ensure the consistency between generated statistic tokens in the next sliding time window and real statistic tokens. Specifically, we calculate $\mathcal{L}_{con}^{t,w}$ for attention module $\mathcal{A}_w$ as

$$\min_{\mathcal{A}_w} \mathcal{L}_{con}^{t,w} = \frac{1}{w-1} \sum_{k=1}^{w-1} \|\hat{\boldsymbol{s}}_w^{t-w+k} - \boldsymbol{s}^{t-w+k}\|_2, \quad (t \geq w+2) \wedge w \geq 2, \tag{11}$$

where $\hat{\boldsymbol{s}}_w^j$ and $\boldsymbol{s}_w^j$, $j = t-w+1, \cdots, t-1$ are respectively the predicted statistic token for time point $t_j$ using the attention module $\mathcal{A}_w$ and the real statistic token from memory pool $\mathcal{M}$. This loss is expected to also capture the evolutionary pattern within the time window.

The fourth loss $\mathcal{L}_{adv}^t$ is introduced with this consideration that the feature extractor $\mathcal{E}$ may overfit to current domain and lacks generalizability. To avoid this, we additionally conduct adversarial training between the feature encoder $\mathcal{E}$ and a discriminator $\mathfrak{D}$. Nevertheless, historical data are unavailable. So we instead resort to the learned/generated statistics at previous time points. Concretely, we randomly sample a batch of $B$ samples $\{\boldsymbol{f'}_1^m, \cdots, \boldsymbol{f'}_B^m\}$ from each distribution $\mathcal{N}(\boldsymbol{\mu}^m, (\exp(\boldsymbol{a}^m))^2)$, $m = 1, 2, \cdots, t-1$, via Eq. 12 at each iteration, and use them as the proxy of historical domains.

$$\boldsymbol{f'}_i^m = \boldsymbol{\mu}^m + \epsilon \cdot \exp(\boldsymbol{a}^m), \quad \epsilon \sim \mathcal{N}(0,1) \wedge -\alpha \leq \epsilon \leq \alpha, \tag{12}$$

where $\alpha$ is a truncation hyper-parameter to control the sampling area. Then discriminator $\mathfrak{D}$ is trained to distinguish $\{\boldsymbol{f}_i^t\}_{i=1}^{N^t}$ from $\{\boldsymbol{f'}_1^m, \cdots, \boldsymbol{f'}_B^m\}_{m=1}^{t-1}$ and the feature extractor tries to confuse $\mathfrak{D}$. Such adversarial process is achieved by the following loss $\mathcal{L}_{adv}^t$:

$$\max_{\mathcal{E}} \min_{\mathfrak{D}} \mathcal{L}_{adv}^t = \frac{1}{2}\left(\frac{1}{B \times (t-1)} \sum_{m=1}^{t-1} \sum_{j=1}^{B} -\log(\mathfrak{D}(\boldsymbol{f'}_j^m)) + \frac{1}{N^t} \sum_{j=1}^{N^t} -\log(1 - \mathfrak{D}(\boldsymbol{f}_j^t))\right), \quad t \geq 2. \tag{13}$$

In practice, the gradient reverse layer (GRL) [15] is used to achieve the adversarial training. To sum up, at time point $t_t$, the model is trained to minimize the following total loss $\mathcal{L}_{total}^t$:

$$\mathcal{L}_{total}^t = \mathcal{L}_{ce}^t + \mathcal{L}_{stan}^t + \sum_{w=1}^{W} \mathcal{L}_{con}^{t,w} + \lambda \mathcal{L}_{adv}^t, \tag{14}$$

Table 1: Accuracy (%) on Yearbook, RMNIST and fMoW. The best and second-best results in CDGTD setup are bolded and underlined. (Yearbook: $K = 5$, RMNIST: $K = 3$, fMoW: $K = 3$)

| Method | Conference | Incremental training | Access multiple domains | Yearbook Accuracy (%) ↑ | | | RMNIST Accuracy (%) ↑ | | | fMoW Accuracy (%) ↑ | | |
|---|---|---|---|---|---|---|---|---|---|---|---|---|
| | | | | $\mathcal{D}^{T+1}$ | OOD avg. | OOD worst | $\mathcal{D}^{T+1}$ | OOD avg. | OOD worst | $\mathcal{D}^{T+1}$ | OOD avg. | OOD worst |
| Offline | - | ✗ | ✓ | 89.30 | 88.46 | 86.81 | 98.15 | 92.14 | 83.89 | 72.43 | 59.76 | 49.85 |
| IRM [2] | arXiv'19 | ✗ | ✓ | 97.09 | 94.52 | 92.58 | 95.10 | 85.05 | 72.52 | 64.77 | 54.92 | 46.51 |
| CORAL [50] | ECCV Workshops'16 | ✗ | ✓ | 95.94 | 91.79 | 88.84 | 93.04 | 79.10 | 62.96 | 62.14 | 51.42 | 42.19 |
| Mixup [61] | ICLR'18 | ✗ | ✓ | 94.98 | 91.12 | 88.35 | 97.11 | 89.66 | 79.63 | 70.27 | 57.73 | 48.04 |
| LISA [57] | ICML'22 | ✗ | ✓ | 95.51 | 92.97 | 91.29 | 96.21 | 87.04 | 75.15 | 70.05 | 55.52 | 44.61 |
| CDOT [43] | arXiv'19 | ✗ | ✓ | 95.17 | 92.90 | 91.46 | 97.96 | 90.19 | 79.67 | - | - | - |
| CIDA [55] | ICML'20 | ✗ | ✓ | 92.36 | 90.67 | 88.45 | 97.43 | 89.19 | 78.32 | - | - | - |
| GI [41] | NeurIPS'21 | ✗ | ✓ | 97.42 | 96.37 | 95.73 | 97.78 | 91.00 | 82.46 | 61.62 | 50.83 | 42.78 |
| LSSAE [44] | ICML'22 | ✗ | ✓ | 93.93 | 92.12 | 88.75 | 96.73 | 90.36 | 82.13 | 59.15 | 48.66 | 41.38 |
| IncFinetune | - | ✓ | ✗ | 96.61 | 94.72 | 93.48 | _98.62_ | 92.80 | 84.61 | 65.52 | 53.99 | 45.23 |
| Mixup [61] | ICLR'18 | ✓ | ✗ | 90.21 | 89.83 | 88.43 | 98.43 | 92.38 | 83.45 | 64.84 | 52.00 | 42.54 |
| SimCLR [11] | ICML'20 | ✓ | ✗ | 95.94 | 93.07 | 89.65 | 98.23 | 90.98 | 81.05 | 64.97 | 53.20 | 44.71 |
| SwAV [8] | NeurIPS'20 | ✓ | ✗ | **97.37** | 94.27 | 91.44 | 98.08 | 90.85 | 80.96 | 66.47 | 54.51 | 45.29 |
| EWC [22] | arXiv'16 | ✓ | ✗ | _97.18_ | _95.12_ | 93.64 | 98.56 | 92.02 | 82.80 | 66.23 | 54.55 | 45.80 |
| SI [60] | ICML'17 | ✓ | ✗ | 97.09 | 94.67 | 93.48 | 98.61 | _93.27_ | 85.65 | 66.61 | _54.89_ | **46.46** |
| A-GEM [10] | ICLR'19 | ✓ | ✗ | 94.36 | 90.96 | 88.88 | 95.99 | 86.95 | 75.45 | 54.54 | 47.61 | 41.13 |
| SGP [47] | AAAI'23 | ✓ | ✗ | 95.65 | 92.92 | 91.39 | 97.12 | 88.97 | 78.05 | - | - | - |
| DRAIN [3] | ICLR'23 | ✓ | ✗ | 96.23 | 94.71 | _93.73_ | 98.52 | 93.09 | _85.75_ | **67.22** | **55.05** | _46.24_ |
| **EvoS** | - | ✓ | ✗ | **97.37** | **95.53** | **94.78** | **98.64** | **93.84** | **87.04** | _67.18_ | 54.64 | 45.86 |

For fMoW, backbone DenseNet-121 is too big to apply full GI and DRAIN. So we apply DRAIN only to the classifier and apply GI without the fine-tuning stage.

where $\lambda$ is a hyper-parameter to balance the loss tradeoff. Once the training procedure at time point $t_t$ is finished, we store the learned/generated statistics of current domain $\mathcal{D}^t$ into memory pool $\mathcal{M}$ by

$$\boldsymbol{\mu}^t, \boldsymbol{a}^t \leftarrow \hat{\boldsymbol{\mu}}^t, \hat{\boldsymbol{a}}^t. \tag{15}$$

In the inference stage, we use MSAM to generate future statistics $\hat{s}^{T+k}$ based on the statistics $\{s^t\}_{t=1}^{T+k-1}$ in memory pool $\mathcal{M}$ and store it into $\mathcal{M}$ as Eq. 15 for the generation at next time point. Due to space limitation, the training and testing procedures are provided in the appendix.

## 4 Experiments

### 4.1 Experimental Setup

Thanks to the work in [56], several real-world datasets with distribution shifts over time have been available. And we evaluate EvoS on three image classification datasets (**Yearbook** and **fMoW** from [56] and **RMNIST**) and two text classification datasets (**Huffpost** and **Arxiv** from [56]). Yearbook collects data from 1930 to 2013, where we treat every four years as a domain and use the first 16 domains for training ($T = 16$), the last 5 domains for testing ($K = 5$). fMoW includes data of 16 years and we set $T = 13, K = 3$. RMNIST contains 9 domains with $T = 6, K = 3$. Huffpost includes data of 7 years with $T = 4, K = 3$. Arxiv collects data of 172 categories for 16 years, with $T = 9, K = 7$. For each training domain of all datasets, we randomly select 90% data as training split and 10% data as validation split. Following the backbones in [56], we use a 4-layer convolutional network [56] for Yearbook, DenseNet-121 [18] pretrained on ImageNet for fMoW, pretrained DistilBERT base model [48] for Huffpost and Arxiv, and the ConvNet in [44] for RMNIST. For each attention module $\mathcal{A}_w$ in MSAM, its dimension of head $d_h$ is set to 8 and its number of heads is set to $w \cdot n_h$. Specifically, $n_h$ is set to 16 for Yearbook, 32 for RMNIST, 64 for fMoW and 128 for Huffpost and Arxiv. For optimization, we use the Adam optimizer with $lr = 1e - 3$ for Yearbook and RMNIST, $lr = 2e - 4$ for fMoW and $lr = 2e - 5$ for Huffpost and Arxiv. The batch size is set to 64 for all datasets. As for hyper-parameters, we select them via grid search using the validation splits of training domains and finally use $\alpha = 2.0$ for RMNIST, $\alpha = 1.0$ for others, $\lambda = 1.0, W = 3$ for all datasets. For all tasks, we report the mean of 3 random trials. Due to space limitations, please refer to appendix for more details. Code is available at https://github.com/BIT-DA/EvoS.

### 4.2 Main Results

In addition to the incremental training scenario, we also provide results in non-incremental scenario with all source domains simultaneously available, which serves as upper bounds. Among compared baselines, "Offline" denotes merging all source domains into one domain and training the model with the merged domain, while "IncFinetune" represents incrementally training the model in a domain-by-domain fashion. For each dataset, we report the accuracy on the nearest target domain ($\mathcal{D}^{T+1}$),

Table 2: Accuracy (%) on Huffpost and Arxiv. The best and second-best results in CDGTD setup are bolded and underlined. (Huffpost: $K = 3$, Axriv: $K = 7$)

| Method | Conference | Incremental training | Access multiple domains | Huffpost Accuracy (%) ↑ | | | Arxiv Accuracy (%) ↑ | | |
|---|---|---|---|---|---|---|---|---|---|
| | | | | $\mathcal{D}^{T+1}$ | OOD avg. | OOD worst | $\mathcal{D}^{T+1}$ | OOD avg. | OOD worst |
| Offline | - | ✗ | ✓ | 72.74 | 71.50 | 69.63 | 57.49 | 52.38 | 49.28 |
| IRM [2] | arXiv'19 | ✗ | ✓ | 71.04 | 70.31 | 68.97 | 51.11 | 45.89 | 42.86 |
| CORAL [50] | ECCV Workshops'16 | ✗ | ✓ | 71.34 | 70.08 | 68.68 | 50.98 | 45.77 | 42.71 |
| Mixup [61] | ICLR'18 | ✗ | ✓ | 73.34 | 71.16 | 69.29 | 57.58 | 52.77 | 49.62 |
| LISA [57] | ICML'22 | ✗ | ✓ | 72.19 | 70.24 | 68.60 | 56.53 | 52.41 | 49.67 |
| GI [41] | NeurIPS'21 | ✗ | ✓ | 68.06 | 66.32 | 64.64 | 53.43 | 49.19 | 46.13 |
| IncFinetune | - | ✓ | ✗ | 73.57 | 71.98 | 69.80 | 56.22 | 52.43 | 49.37 |
| Mixup [61] | ICLR'18 | ✓ | ✗ | 73.07 | 71.52 | 69.44 | **56.64** | 52.95 | 49.97 |
| EWC [22] | arXiv'16 | ✓ | ✗ | **73.64** | 71.53 | 68.99 | 56.60 | 52.78 | 49.73 |
| SI [60] | ICML'17 | ✓ | ✗ | 72.58 | 71.50 | 69.61 | 49.98 | 47.27 | 44.77 |
| A-GEM [10] | ICLR'19 | ✓ | ✗ | 72.23 | 71.16 | 69.10 | 52.02 | 48.91 | 46.03 |
| DRAIN [3] | ICLR'23 | ✓ | ✗ | 73.42 | 71.75 | 69.69 | 56.04 | 52.07 | 48.97 |
| **EvoS** | - | ✓ | ✗ | 73.42 | **72.36** | **70.19** | 56.60 | **53.15** | **50.19** |

For Huffpost and Arxiv, backbone DistilBERT-base is too big to apply the full GI and DRAIN. So we apply DRAIN only to the classifier and apply GI without the fine-tuning stage.

the average and worst accuracy of future $K$ domains ("OOD avg.": $\frac{1}{K}\sum_{k=1}^{K} Acc(\mathcal{D}^{T+k})$, "OOD worst": $\min_{k\in\{1,2,\cdots,K\}} Acc(\mathcal{D}^{T+k})$).

**Results on image classification tasks** are provided in Table 5, where we can observe that EvoS consistently achieves superior performance across three reported metrics on Yearbook and RMNIST datasets, compared with the methods in incremental training scenario. Especially, EvoS outperforms DRAIN by over 1% according the metric "OOD worst" on Yearbook and RMNIST, which means that our EvoS learns more robust evolving patterns. In addition, we notice that conventional DG methods IRM [2] and CORAL [50] perform worse than temporal DG method GI [41] on Yearbook and RMNIST, showing the importance of leveraging evolutionary patterns for generalization over temporal drifts. Though being effective for the small networks used in Yearbook and RMNIST, the advantage of GI disappears on fMoW dataset with backbone DenseNet-121, where its first-order Taylor expansion requires extremely huge computing resources (over 80G). And our available resources cannot afford it. When ablating the Taylor expansion, the performance of GI on fMoW is not the best. So GI is hard to expand on larger models and the similar issue also exists in DRAIN and LSSAE. Finally, we find that for dataset fMoW, not only our method performs unsatisfactorily but also other methods, except for the "offline" method (i.e., merging all source domain into one domain to train the model). We infer this may be due to the temporal drift is not well presented in this dataset.

**Results on text classification tasks** are shown in Table 6, where EvoS consistently achieves the best performances according to the average and worst accuracy of future $K$ domains under the CDGTD setting. This can be owed to the more robust evolving patterns captured by our multi-scale attention module (MSAM). Similarly, IRM and CORAL show obvious performance drop compared with DRAIN, once again demonstrating the necessity to learn evolving patterns for the problem of CDGTD. And the large performance gap between continual learning methods (i.e., SI and A-GEM) and our method EvoS on dataset Arxiv verifies that our method EvoS fully utilizes the historical knowledge to learn evolutionary patterns, while SI and A-GEM do not consider this.

## 4.3 Analytical Experiments

**Ablation Study.** Firstly, we study the influence of the stopping gradient sg(·) in the losses $\mathcal{L}_{ce}^{t}$ and $\mathcal{L}_{stan}^{t}$. The results of variant A and B in Table 3 almost degenerate to random predictions, compared with variant C. This suggests that using sg(·) in loss $\mathcal{L}_{stan}^{t}$ is essential. Otherwise, the gradient of $\mathcal{L}_{stan}^{t}$ would largely distort the learning of feature encoder, causing training collapse. Meanwhile, the inferior performance of variant C to EvoS also indicates that the gradient of $\mathcal{L}_{ce}^{t}$ should not interfere with the learning of MSAM. Secondly, we ablate the losses $\mathcal{L}_{adv}^{t}$ and $\mathcal{L}_{con}^{t,w}$ to testify their necessity. We can see that the results of

Table 3: Ablation study of EvoS on dataset Yearbook.

| Method | Scale | | Loss | | Using sg(·)? | | Using | OOD |
|---|---|---|---|---|---|---|---|---|
| | single | multi | $\mathcal{L}_{adv}^{t}$ | $\mathcal{L}_{con}^{t,w}$ | $\mathcal{L}_{ce}^{t}$ | $\mathcal{L}_{stan}^{t}$ | truncation $\alpha$? | avg. |
| Variant A | - | ✓ | ✓ | ✓ | No | No | Yes | 51.28 |
| Variant B | - | ✓ | ✓ | ✓ | Yes | No | Yes | 55.37 |
| Variant C | - | ✓ | ✓ | ✓ | No | Yes | Yes | 93.97 |
| Variant D | - | ✓ | - | - | Yes | Yes | Yes | 93.25 |
| Variant E | - | ✓ | ✓ | - | Yes | Yes | Yes | 94.75 |
| Variant F | - | ✓ | - | ✓ | Yes | Yes | Yes | 94.27 |
| Variant G | - | ✓ | ✓ | ✓ | Yes | Yes | No | 94.46 |
| EvoS | - | ✓ | ✓ | ✓ | Yes | Yes | Yes | **95.53** |
| Variant H | ✓ | - | ✓ | ✓ | Yes | Yes | Yes | 94.09 |

Table 4: Results of using historical domain distributions in different ways during adversarial training.

| Method | Used historical domain distribution in $\mathcal{L}_{adv}^t$ | Yearbook Accuracy (%) ↑ | | | RMNIST Accuracy (%) ↑ | | | fMoW Accuracy (%) ↑ | | | Huffpost Accuracy (%) ↑ | | | Arxiv Accuracy (%) ↑ | | |
|---|---|---|---|---|---|---|---|---|---|---|---|---|---|---|---|---|
| | | $\mathcal{D}^{T+1}$ | OOD avg. | OOD worst | $\mathcal{D}^{T+1}$ | OOD avg. | OOD worst | $\mathcal{D}^{T+1}$ | OOD avg. | OOD worst | $\mathcal{D}^{T+1}$ | OOD avg. | OOD worst | $\mathcal{D}^{T+1}$ | OOD avg. | OOD worst |
| EvoS† | random one | 96.56 | 95.40 | 94.46 | 98.05 | 93.04 | 85.46 | 66.62 | 53.81 | 44.76 | 72.99 | 71.57 | 69.14 | 55.89 | 52.45 | 49.52 |
| EvoS | all historical | **97.37** | **95.53** | **94.78** | **98.64** | **93.84** | **87.04** | **67.18** | **54.64** | **45.86** | **73.42** | **72.36** | **70.19** | **56.60** | **53.15** | **50.19** |

EvoS† denotes that we randomly select a domain distribution from the memory pool $\mathcal{M}$ to sample a batch of $B$ features for participating in the adversarial training.

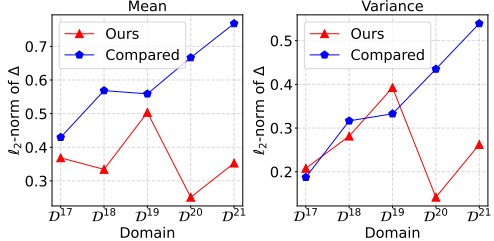

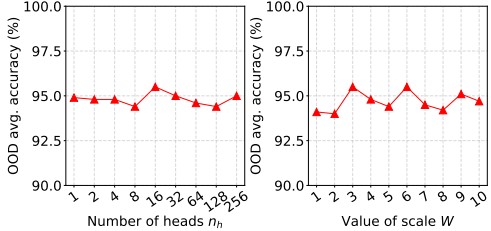

(a) Discrepancy between real and generated statistics.    (b) Effect of the number of heads and scale.

Figure 2: (a): The $l$-2 norm of the difference $\Delta$ between the statistics calculated on each test domain and the statistics generated by MSAM on Yearbook. "Ours" refers to the way in Eq. 10 and "Compared" refers to minimizing the $l$-2 norm of $\Delta$ between statistics calculated in each batch and that generated by MSAM. (b): Effects of using different numbers of heads and scales on Yearbook.

variant D, E and F are all inferior to EvoS. Such results verify two points. One is that the adversarial training conduces to better generalization and mitigating the problem of overfitting. And the other is that the consistency between generated statistic tokens in the window and real statistic tokens in the memory pool imposes stronger constraint to attention module $\mathcal{A}_w$, helping to model the evolving pattern more accurately. Thirdly, we testify the influence of truncation when sampling features from normal distributions. The worse result of variant G than EvoS may be due to the sampled outliers hinder the learning of a shared feature space. Finally, the results of variant H and EvoS compare single-scale and multi-scale attention. The better result of EvoS shows the superiority of MSAM.

Last but not least, in Table 4, we investigate the effect of randomly selecting a historical domain distribution *vs* using all historical domain distributions in the adversarial training. From the results, we see that randomly selecting one historical domain distribution performs worse. This may be because the feature space to be aligned frequently changes if using this manner, making the optimization challenging. By contrast, it is more stable to simultaneously leverage all the preserved domain distributions in the memory pool $\mathcal{M}$ in each iteration for the adversarial training.

**Choices of $\mathcal{L}_{stan}^t$.** In addition to training $\mathcal{A}_w$ via loss $\mathcal{L}_{stan}^t$ in Eq. 10, we also try another way. Concretely, we directly minimize the $l$-2 norm of the difference between statistics calculated in each batch and that generated by MSAM, i.e., $\mathcal{L}_{stan}^t = \|\frac{1}{B}\sum_{i=1}^{B}\mathrm{sg}(\boldsymbol{f}_i^t) - \hat{\boldsymbol{\mu}}^t\|_2 + \|\frac{1}{B-1}\sum_{i=1}^{B}(\mathrm{sg}(\boldsymbol{f}_i^t) - \frac{1}{B}\sum_{j=1}^{B}\mathrm{sg}(\boldsymbol{f}_j^t))^2 - (\hat{\boldsymbol{\sigma}}^t)^2\|_2$. Fig. 2(a) plots the discrepancy between generated and real statistics on each test domain of Yearbook when using the two ways to train the attention module. We see that the way used in Sec. 3.3 yields a smaller discrepancy between generated and real statistics, implying that the learned evolving pattern is more accurate. So we choose to use it throughout the experiments.

**Effect of Number of Heads and Scale.** We further investigate the effect of number of heads and scale of MSAM in Fig. 2(b). Varying the number of heads, the performance exhibits an inverted V-shaped trend, because too few heads would hamper the learning capability of the attention module while too many heads could lead to overfitting. As for difference scales (i.e., the maximum length of sliding time window), we find that multi-scales ($W \geq 1$) yields better performance than single-scale ($W = 1$), showing the superiority of MSAM. Yet, more scales do not always produce better results. One possible explanation is that an excessively large scale relative to the domain sequence length leads to an inadequate number of sliding windows, hampering the learning of attention module.

**Visualization of Decision Boundary.** In this qualitative experiment, we visualize the change of decison boundary on the inter-twinning moons 2D problem in a gradual manner. Concretely, the 2-Moons dataset in [41] is used, where 10 domains (0 to 9) are obtained by rotating data points counter-clockwise in units of $18°$. In Figure 3, the model is sequentially trained using EvoS until the $t$-th domain is finished, and then we visualize the decision boundary on current domain $\mathcal{D}^t$ and the next future domain $\mathcal{D}^{t+1}$. From the results, we can observe that the decision boundary successfully adapts to future domains, showing that EvoS can truly capture the underlying temporal drift of data.

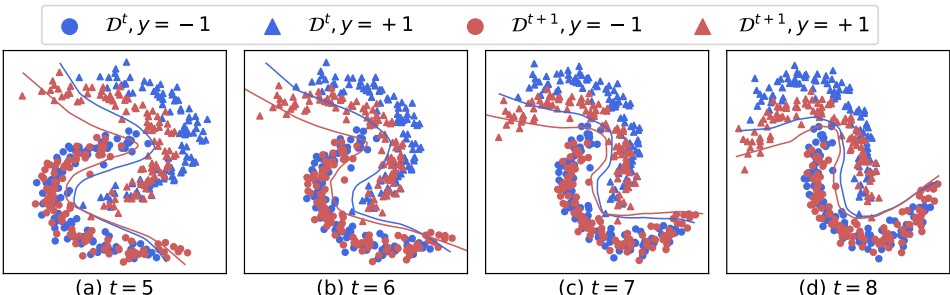

Figure 3: Visualization of decision boundaries for the model at current time point $\mathfrak{t}_t$ on inter-twinning moons 2D problem. Blue line and points are the decision boundary and samples in current domain $\mathcal{D}^t$, and red line and points are the decision boundary and samples in the next future domain $\mathcal{D}^{t+1}$.

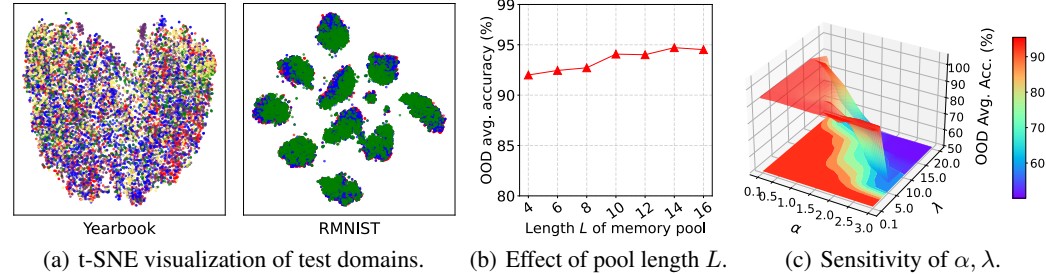

(a) t-SNE visualization of test domains.  (b) Effect of pool length $L$.  (c) Sensitivity of $\alpha, \lambda$.

Figure 4: (a): Visualization of standardized features from each test domain on Yearbook and RMNIST. Different colors denote different domains. (b): Effect of the memory pool length $L$ on dataset Yearbook. (c):Hyper-parameter sensitivity of $\alpha$ and $\lambda$ on Yearbook

**Effect of Memory Pool Length.** In the main experiments, we do not restrict the size of the memory pool $\mathcal{M}$, since the datasets used in our paper has a moderate number of domains and the memory cost is small. Nevertheless, a fixed memory pool size is more practical when considering a lifelong process, i.e., $T \to \infty$. Thus, we additionally conduct the experiment where the memory pool $\mathcal{M}$ is implemented as a FIFO queue with different fixed length $L$. That is, only the statistics for up to the $L$ most recent historical domains can be stored. The results on Yearbook in Fig. 4(b) demonstrate that our method generally performs well and a relatively large memory pool length is better.

**t-SNE Visualization of Standardized Features.** To verify our method appropriate for generalization, we visualize the standardized features of all target domains via t-SNE [38] on RMNIST and Yearbook in Fig. 4(a). It can be observed that the standardized features of target domains are well aligned, suggesting that MASM can capture evolving patterns effectively and feature standardization helps address temporal drift appropriately. These enable EvoS to achieve generalization on future domains.

**Hyper-parameter Sensitivity.** $\alpha$ and $\lambda$ control the truncation range and the tradeoff the adversarial loss $\mathcal{L}_{adv}$, respectively. Fig. 4(c) shows the sensitivity of EvoS to them on Yearbook, where $\alpha \in \{0.1, 0.5, 1.0, 1.5, 2.0, 2.5, 3.0\}$ and $\lambda \in \{0.1, 0.5, 1.0, 1.5, 2.0, 5.0, 10.0, 15.0, 20.0\}$. We see that EvoS is more sensitive to $\lambda$ than $\alpha$, and large $\lambda$ values significantly worsen performance. In practice, we suggest selecting $\lambda$ from a range $\leq 1$. For $\alpha$, its effect is modest and $\alpha = 1$ generally works well.

## 5   Conclusion

This paper considers a more challenging problem of continual domain generalization over temporal drift (CDGTD) than conventional DG, where the model is incrementally trained with sequential domains and is required to generalize to unseen domains that are not too far into the future. To this end, we propose an Evolving Standardization (EvoS) method to learn the evolving pattern of sequential domains over the temporal drift and hope to achieve the generalization by transforming the domain distribution into a common normal distribution via feature standardization. Experiments on real-world datasets including images and texts verify the efficacy of EvoS. Since existing DG works focus on conventional setting, we hope this work can encourage more research works on CDGTD.

## Acknowledgements

This paper was supported by National Key R&D Program of China (No. 2021YFB3301503), the National Natural Science Foundation of China (No. 62376026), and also sponsored by Beijing Nova Program (No. 20230484296).

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

## Appendix Contents

## A  Broader Impacts & Limitations

Our work focus on the problem of continual domain generalization over temporal drift (CDGTD), which aims to generalize the model to unseen future domains by leveraging underlying evolutionary patterns. The effectiveness of our method on several real-world datasets means that it may potentially benefit relevant applications and communities that deal with temporal drifts, e.g., advertisement recommendation, autonomous driving, popularity forecast of media content, etc. Nevertheless, we should also be cautious about possible failures of our method when encountering sudden distribution shifts. In the future, we may explore the automatic identification of domains with severe distribution shifts, allowing us to proactively reject them and mitigate the risk of severe accidents.

## B  Ethics Statement & Licenses

All the datasets used in the paper are publicly available and are only intended to compare the performances of different algorithms on classification tasks, adhering to the following licenses of these datasets:

- Yearbook: MIT licensed
- MNIST: CC BY-SA 3.0
- fMoW: The Functional Map of the World Challenge Public License
- Huffpost: CC0: Public Domain
- arXiv: CC0: Public Domain

## C  Algorithm of EvoS

The training and inference procedure for EvoS is provided in Algorithm 1 and 2, respectively.

## D  Experimental Setup Details

### D.1  Dataset Description

**Yearbook** used in our paper is from [56]. It is based on the Portraits [16] dataset (MIT license), containing the $32 \times 32$ grayscale images of yearbook portraits from 128 American high schools in 27 states. The data spanning from 1930 to 2013 reflects the evolving fashion styles and shifting social norms throughout the decades. The task is the binary classification of genders. Note that it is only used for comparing the generalization performances of different algorithms on classification tasks. For this dataset, we treat every four years as a domain, resulting in 16 domains. Table 11 provides the number of samples in each domain. The first 16 domains are used for training ($T = 16$), the last 5 domains for testing ($K = 5$).

**RMNIST** is a variant of the MNIST [13] dataset, which comprises 9 domains generated by applying the rotations with degree of $0°, 10°, \cdots, 80°$ in order to MNIST to simulate the temporal drift. The task is to classify a $28 \times 28$ grayscale digit image from 0 to 9. We use the first 6 domains for training ($T = 6$) and the last 3 for testing ($K = 3$).

**Algorithm 1:** Training procedure for EvoS

**Input:** sequential labeled training domains $\{\mathcal{D}^1, \mathcal{D}^2, \cdots, \mathcal{D}^T\}$, hyper-parameters $\alpha, \lambda, W$, feature encoder $\mathcal{E}$, classifier $\mathcal{C}$, learnable statistic vectors $\hat{\boldsymbol{\mu}}^1, \hat{\boldsymbol{a}}^1, \hat{\boldsymbol{\mu}}^2, \hat{\boldsymbol{a}}^2$, multi-head self attention modules $\{\mathcal{A}_w\}_{w=1}^W$, memory pool $\mathcal{M}$, batch size $B$, training steps $E$.

1   **for** $t = 1$ **to** $T$ **do**

2     /* - - - - - - - - - - *train on t-th domain* - - - - - - - - - - */

3     **for** $e = 1$ **to** $E$ **do**

4        Randomly sample a batch of data $\{\boldsymbol{x}_i^t, y_i^t\}_{i=1}^B$ from domain $\mathcal{D}^t$.

5        Obtain features $\{\boldsymbol{f}_i^t\}_{i=1}^B$ by feeding $\{\boldsymbol{x}_i^t\}_{i=1}^B$ into the feature encoder $\mathcal{E}$.

6        $\mathcal{L}_{total}^t = 0$.

7        /* - - - - - - - - - - *multi-scale attention module* - - - - - - - - - - */

8        **for** $w = 1$ **to** $W$ **do**

9           **if** $t \geq w + 2$ **then**

10             Prepare the input $\ddot{\boldsymbol{S}}_w^{t-1} = [\ddot{\boldsymbol{s}}_w^1; \ddot{\boldsymbol{s}}_w^2; \cdots; \ddot{\boldsymbol{s}}_w^{t-w}]$ for $\mathcal{A}_w$ by using a sliding time window with length $w$ on the memory pool $\mathcal{M}$, as described in Eq. 4.

11             Obtain the output $\breve{\boldsymbol{S}}_w^t = \mathcal{A}_w(\ddot{\boldsymbol{S}}_w^{t-1}) = [\breve{\boldsymbol{s}}_w^1; \breve{\boldsymbol{s}}_w^2; \cdots; \breve{\boldsymbol{s}}_w^{t-w}]$.

12             Compute the average of output tokens: $\breve{\boldsymbol{s}}_w^{t-w+1} = avg(\breve{\boldsymbol{S}}_w^t) = \frac{1}{t-w} \sum_{i=1}^{t-w} \breve{\boldsymbol{s}}_w^i$.

13             Split $\breve{\boldsymbol{s}}_w^{t-w+1}$ into $w$ parts: $[\hat{\boldsymbol{s}}_w^{t-w+1}, \cdots, \hat{\boldsymbol{s}}_w^{t-1}, \hat{\boldsymbol{s}}_w^t] = \breve{\boldsymbol{s}}_w^{t-w+1}$.

14             Calculate loss $\mathcal{L}_{con}^{t,w}$ by Eq. 11 and $\mathcal{L}_{total}^t += \mathcal{L}_{con}^{t,w}$.

15        /* - - - - - - - - - - *calculate losses* - - - - - - - - - - */

16        **if** $t >= 3$ **then**

17           Generate the statistics for current domain as $[\hat{\boldsymbol{\mu}}^t, \hat{\boldsymbol{a}}^t] = \hat{\boldsymbol{s}}^t = \frac{1}{W} \sum_{w=1}^W \hat{\boldsymbol{s}}_w^t$.

18        Calculate loss $\mathcal{L}_{ce}^t$ by Eq. 8 and $\mathcal{L}_{stan}^t$ by Eq. 10 and $\mathcal{L}_{total}^t += (\mathcal{L}_{ce}^t + \mathcal{L}_{stan}^t)$.

19        **if** $t \geq 2$ **then**

20           Sample features $\{\boldsymbol{f'}_1^m, \cdots, \boldsymbol{f'}_B^m\}_{m=1}^{t-1}$ by Eq. 12 as the proxy of historical domains.

21           Calculate loss $\mathcal{L}_{adv}^t$ by Eq. 13 and $\mathcal{L}_{total}^t += \lambda \mathcal{L}_{adv}^t$.

22        Optimize corresponding modules by minimizing loss $\mathcal{L}_{total}^t$.

23     /* - - - - - - - - - - *store statistics into the memory pool* $\mathcal{M}$ - - - - - - - - - - */

24     **if** $t >= 3$ **then**

25        Generate statistics $\hat{\boldsymbol{\mu}}^t, \hat{\boldsymbol{a}}^t$ by repeating step 8 to 17.

26     Store the learned/generated statistics into the memory pool $\mathcal{M}$ by $\boldsymbol{\mu}^t, \boldsymbol{a}^t \leftarrow \hat{\boldsymbol{\mu}}^t, \hat{\boldsymbol{a}}^t$.

27 **return** *Final* $\mathcal{E}, \mathcal{C}, \{\mathcal{A}_w\}_{w=1}^W, \mathcal{M}$.

**fMoW** used in our paper is from [56]. It collects the $224 \times 224$ RGB satellite images from 2002 to 2017 over 200 countries, where the visual features present in satellite data undergo changes over time due to both human and environmental activities. The task to classify the functional purpose of the buildings or the land in the images into one of 62 categories. For this dataset, we treat every year as a domain and Table 12 provides the number of samples in each domain. The first 13 domains are used for training ($T = 13$) and the last 3 domains are used for testing ($K = 3$).

**Huffpost** in [56] comprises news headlines from the Huffington Post from 2012 to 2018, the task of which is to classify the news headline into one of 11 news categories ("Black Voices", "Business", "Comedy", "Crime", "Entertainment", "Impact", "Queer Voices", "Science", "Sports", "Tech", "Travel"). The data spanning for 2012 to 2018 presents changes in the content or style of news along the time dimension. For this dataset, we use the first 4 years for training ($T = 4$) and the last 3 years for testing ($K = 3$). The number of samples in each domain is provided in Table 13.

**Arxiv** in [56] contains paper titles and their corresponding primary categories spanning from 2007 to 2022. The content of Arxiv preprints evolves over time, reflecting the dynamic nature of research fields. And the task is to classify a research paper into one of 172 categories based on its title. We use the data from the first 9 domains for training ($T = 9$) and the data from the last 7 years for testing ($K = 7$). The number of samples in each domain is presented in Table 14.

---

**Algorithm 2:** Inference procedure for EvoS

---

**Input:** sequential target domains $\{\mathcal{D}^{T+1}, \mathcal{D}^{T+2}, \cdots, \mathcal{D}^{T+K}\}$, feature encoder $\mathcal{E}$, classifier $\mathcal{C}$, multi-head self attention modules $\{\mathcal{A}_w\}_{w=1}^{W}$, memory pool $\mathcal{M}$.

1 **for** $t = T + 1$ **to** $T + K$ **do**

2     /* - - - - - - - - - *generate statistics via multi-scale attention module* - - - - - - - - - */

3     **for** $w = 1$ **to** $W$ **do**

4         Prepare the input $\ddot{\boldsymbol{S}}_w^{t-1} = [\ddot{\boldsymbol{s}}_w^1; \ddot{\boldsymbol{s}}_w^2; \cdots; \ddot{\boldsymbol{s}}_w^{t-w}]$ for $\mathcal{A}_w$ by using a sliding time window with length $w$ on the memory pool $\mathcal{M}$, as described in Eq. 4.

5         Obtain the output $\breve{\boldsymbol{S}}_w^t = \mathcal{A}_w(\ddot{\boldsymbol{S}}_w^{t-1}) = [\breve{\boldsymbol{s}}_w^1; \breve{\boldsymbol{s}}_w^2; \cdots; \breve{\boldsymbol{s}}_w^{t-w}]$.

6         Compute the average of output tokens: $\breve{\boldsymbol{s}}_w^{t-w+1} = avg(\breve{\boldsymbol{S}}_w^t) = \frac{1}{t-w}\sum_{i=1}^{t-w} \breve{\boldsymbol{s}}_w^i$.

7         Split $\breve{\boldsymbol{s}}_w^{t-w+1}$ into $w$ parts: $[\hat{\boldsymbol{s}}_w^{t-w+1}, \cdots, \hat{\boldsymbol{s}}_w^{t-1}, \hat{\boldsymbol{s}}_w^t] = \breve{\boldsymbol{s}}_w^{t-w+1}$.

8     Generate the statistics for current domain as $[\hat{\boldsymbol{\mu}}^t, \hat{\boldsymbol{a}}^t] = \hat{\boldsymbol{s}}^t = \frac{1}{W}\sum_{w=1}^{W} \hat{\boldsymbol{s}}_w^t$.

9     /* - - - - - - - - - - *inference on t-th domain* - - - - - - - - - - */

10     **for** $\boldsymbol{x}^t \in \mathcal{D}^t$ **do**

11         Obtain features $\boldsymbol{f}^t$ by feeding $\boldsymbol{x}^t$ into the feature encoder $\mathcal{E}$.

12         Conduct feature standardization by $\boldsymbol{z}^t = \frac{\boldsymbol{f}^t - \hat{\boldsymbol{\mu}}^t}{\hat{\boldsymbol{\sigma}}^t} = \frac{\boldsymbol{f}^t - \hat{\boldsymbol{\mu}}^t}{\exp(\hat{\boldsymbol{a}}^t)}$.

13         Generate the prediction $\hat{y}^t$ by feeding $\boldsymbol{z}^t$ into the classifier $\mathcal{C}$.

14     /* - - - - - - - - - - *store statistics into the memory pool* $\mathcal{M}$ - - - - - - - - - - */

15     Store the generated statistics into the memory pool $\mathcal{M}$ by $\boldsymbol{\mu}^t, \boldsymbol{a}^t \leftarrow \hat{\boldsymbol{\mu}}^t, \hat{\boldsymbol{a}}^t$.

---

For each training domain of all datasets, we randomly select 90% of data as training split and 10% of data as validation split. The evaluation of generalization performance is based on the whole data of each test domain.

### D.2 Implementation Details

All experiments are implemented via PyTorch and the backbones we use mainly adhere to [56]. **Yearbook** uses a 4-layer convolutional network from [56] as the backbone. For multi-scale attention module (MSAM), we set the dimension of head as $d_h = 8$ for each attention module $\mathcal{A}_w$ and the number of heads for $\mathcal{A}_w$ as $w \cdot n_h$, with $n_h = 16, w = 1, \cdots, W$. And the Adam optimizer with $lr = 1e-3$ is used to optimize the model, where the batch size $B$ is set to 64 and the training epochs of each domain are set to 50. As for the hyper-parameters, we use $\alpha = 1.0, \lambda = 1.0, W = 3$.

**RMNIST** adopts the ConvNet in [44] as the backbone, and we set the dimension of head as $d_h = 8$ for each attention module $\mathcal{A}_w$ and the number of heads for $\mathcal{A}_w$ as $w \cdot n_h$, with $n_h = 32, w = 1, \cdots, W$. The Adam optimizer with $lr = 1e-3$ is adopted for model optimization. The batch size and training epochs of each domain are set to 64 and 50, respectively. And $\alpha = 2.0, \lambda = 1.0, W = 3$ is used.

**fMoW** employ the DenseNet-121 [18] pretrained on ImageNet as the backbone. Besides, we use a bottleneck layer [28] to reduce the feature dimensions into 256, and set the dimension of head as $d_h = 8$ for each attention module $\mathcal{A}_w$ and the number of heads for $\mathcal{A}_w$ as $w \cdot n_h$, with $n_h = 64, w = 1, \cdots, W$. Similarly, the Adam optimizer is used, where the learning rate is set to $lr = 2e-4$. The batch size is set to $B = 64$ and each training domain is trained for 25 epochs. As for the hyper-parameters, we set $\alpha = 1.0, \lambda = 1.0, W = 3$ for fMoW dataset.

**Huffpost** and **Arxiv** use the pretrained DistilBERT base model [48] as the backbone. For MSAM, we set the dimension of head as $d_h = 8$ for each attention module $\mathcal{A}_w$ and the number of heads for $\mathcal{A}_w$ as $w \cdot n_h$, with $n_h = 128, w = 1, \cdots, W$. Also the Adam optimizer is used with the learning rate $lr = 2e-5$ and batch size $B = 64$. And $\alpha = 1.0, \lambda = 1.0, W = 3$ is used for the two dataset. For the Huffpost dataset, each training is trained for 50 epochs, while the training epochs of each domain are set to 5 for the Arxiv dataset.

We run each task on a single NVIDIA GeForce RTX 3090 GPU for three random trials. For baselines, we also select their hyper-parameters via the grid search using the validation splits of training domains.

Table 5: Accuracy (%) on Yearbook, RMNIST and fMoW. The best and second-best results in CDGTD setup are bolded and underlined. (Yearbook: $K = 5$, RMNIST: $K = 3$, fMoW: $K = 3$)

| Method | Incremental training | Access multiple domains | Yearbook Accuracy (%) ↑ | | | RMNIST Accuracy (%) ↑ | | | fMoW Accuracy (%) ↑ | | |
|---|---|---|---|---|---|---|---|---|---|---|---|
| | | | $\mathcal{D}^{T+1}$ | OOD avg. | OOD worst | $\mathcal{D}^{T+1}$ | OOD avg. | OOD worst | $\mathcal{D}^{T+1}$ | OOD avg. | OOD worst |
| Offline | ✗ | ✓ | 89.30 (3.15) | 88.46 (2.72) | 86.81 (3.06) | 98.15 (0.41) | 92.14 (0.91) | 83.89 (1.38) | 72.43 (1.31) | 59.76 (1.91) | 49.85 (3.04) |
| IRM [2] | ✗ | ✓ | 97.09 (0.38) | 94.52 (0.41) | 92.58 (0.52) | 95.10 (1.98) | 85.05 (3.17) | 72.52 (3.81) | 64.77 (1.57) | 54.92 (2.16) | 46.51 (3.47) |
| CORAL [50] | ✗ | ✓ | 95.94 (1.43) | 91.79 (3.04) | 88.84 (6.71) | 93.04 (0.35) | 79.10 (1.09) | 62.96 (1.70) | 62.14 (1.32) | 51.42 (2.13) | 42.19 (3.07) |
| Mixup [61] | ✗ | ✓ | 94.98 (1.84) | 91.12 (1.92) | 88.35 (3.45) | 97.11 (0.41) | 89.66 (0.99) | 79.63 (1.47) | 70.27 (0.91) | 57.73 (1.15) | 48.04 (1.43) |
| LISA [57] | ✗ | ✓ | 95.51 (0.78) | 92.97 (0.57) | 91.29 (0.42) | 96.21 (0.51) | 87.04 (1.94) | 75.15 (2.43) | 70.05 (0.64) | 55.52 (0.96) | 44.61 (1.02) |
| CDOT [43] | ✗ | ✓ | 95.17 (1.98) | 92.90 (1.81) | 91.46 (2.04) | 97.96 (1.00) | 90.19 (0.93) | 79.67 (1.27) | - | - | - |
| CIDA [55] | ✗ | ✓ | 92.36 (1.22) | 90.67 (1.45) | 88.45 (1.69) | 97.43 (1.08) | 89.19 (0.89) | 78.32 (1.46) | - | - | - |
| GI [41] | ✗ | ✓ | 97.42 (0.32) | 96.37 (0.33) | 95.73 (0.52) | 97.78 (0.04) | 91.00 (0.70) | 82.46 (2.01) | 61.62 (1.27) | 50.83 (2.19) | 42.78 (2.57) |
| LSSAE [44] | ✗ | ✓ | 93.93 (1.91) | 92.12 (2.06) | 88.75 (4.11) | 96.73 (0.31) | 90.36 (0.46) | 82.13 (0.71) | 59.15 (1.25) | 48.66 (1.90) | 41.38 (2.49) |
| IncFinetune | ✓ | ✗ | 96.61 (0.17) | 94.72 (0.11) | 93.48 (0.54) | 98.62 (0.22) | 92.80 (0.65) | 84.61 (1.15) | 65.52 (0.34) | 53.99 (0.46) | 45.23 (0.83) |
| Mixup [61] | ✓ | ✗ | 90.21 (1.92) | 89.83 (1.58) | 88.43 (1.67) | 98.43 (0.12) | 90.49 (0.37) | 81.05 (1.12) | 64.84 (0.23) | 52.00 (0.37) | 44.71 (0.63) |
| SimCLR [11] | ✓ | ✗ | 95.94 (0.55) | 93.07 (0.67) | 89.65 (1.70) | 98.23 (0.24) | 90.98 (0.61) | 81.05 (1.12) | 64.97 (0.36) | 53.20 (0.59) | 44.71 (0.63) |
| SwAV [8] | ✓ | ✗ | **97.37 (0.12)** | 94.27 (0.07) | 91.44 (2.31) | 98.08 (0.09) | 90.85 (0.38) | 80.96 (0.85) | 66.47 (0.10) | 54.51 (0.18) | 45.29 (0.21) |
| EWC [22] | ✓ | ✗ | 97.18 (0.12) | 95.12 (0.07) | 93.64 (0.37) | 98.56 (0.08) | 92.02 (0.31) | 82.80 (0.67) | 66.23 (0.12) | 54.55 (0.16) | 45.80 (0.18) |
| SI [60] | ✓ | ✗ | 97.09 (0.09) | 94.67 (0.11) | 93.48 (0.21) | 98.61 (0.04) | 93.27 (0.15) | 85.65 (1.21) | 66.61 (0.05) | 54.89 (0.11) | **46.46 (0.15)** |
| A-GEM [10] | ✓ | ✗ | 94.36 (0.69) | 90.96 (0.51) | 88.88 (0.29) | 95.99 (0.66) | 86.95 (1.63) | 75.45 (2.59) | 54.54 (1.98) | 47.61 (2.46) | 41.13 (3.57) |
| SGP [47] | ✓ | ✗ | 95.65 (0.58) | 92.92 (0.45) | 91.39 (0.22) | 97.12 (0.37) | 88.97 (0.50) | 78.05 (0.95) | - | - | - |
| DRAIN [3] | ✓ | ✗ | 96.23 (0.33) | 94.71 (0.45) | 93.73 (0.64) | 98.52 (0.07) | 91.09 (0.14) | 85.75 (0.24) | **67.22 (0.04)** | **55.05 (0.09)** | 46.24 (0.12) |
| **EvoS** | ✓ | ✗ | **97.37 (0.03)** | **95.53 (0.36)** | **94.78 (0.46)** | **98.64 (0.02)** | **93.84 (0.16)** | **87.04 (0.36)** | 67.18 (0.05) | 54.64 (0.11) | 45.86 (0.21) |

For fMoW, backbone DenseNet-121 is too big to apply full GI and DRAIN. So we apply DRAIN only to the classifier and apply GI without the fine-tuning stage.

Table 6: Accuracy (%) on Huffpost and Arxiv. The best and second-best results in CDGTD setup are bolded and underlined. (Huffpost: $K = 3$, Axriv: $K = 7$)

| Method | Incremental training | Access multiple domains | Huffpost Accuracy (%) ↑ | | | Arxiv Accuracy (%) ↑ | | |
|---|---|---|---|---|---|---|---|---|
| | | | $\mathcal{D}^{T+1}$ | OOD avg. | OOD worst | $\mathcal{D}^{T+1}$ | OOD avg. | OOD worst |
| Offline | ✗ | ✓ | 72.74 (0.14) | 71.50 (0.56) | 69.63 (0.81) | 57.49 (0.15) | 52.38 (0.43) | 49.28 (0.67) |
| IRM [2] | ✗ | ✓ | 71.04 (0.45) | 70.31 (0.67) | 68.97 (0.87) | 51.11 (1.04) | 45.89 (2.77) | 42.86 (3.98) |
| CORAL [50] | ✗ | ✓ | 71.34 (0.51) | 70.08 (0.69) | 68.68 (0.94) | 50.98 (1.34) | 45.77 (2.92) | 42.71 (4.15) |
| Mixup [61] | ✗ | ✓ | 73.34 (0.02) | 71.16 (0.07) | 69.29 (0.12) | 57.58 (0.03) | 52.77 (0.19) | 49.62 (0.24) |
| LISA [57] | ✗ | ✓ | 72.19 (0.06) | 70.24 (0.56) | 68.60 (0.91) | 56.53 (0.02) | 52.41 (0.11) | 49.67 (0.23) |
| GI [41] | ✗ | ✓ | 68.06 (1.51) | 66.32 (2.78) | 64.64 (3.67) | 53.43 (1.65) | 49.19 (2.49) | 46.13 (3.06) |
| IncFinetune | ✓ | ✗ | 73.57 (0.02) | 71.98 (0.06) | 69.80 (0.11) | 56.22 (0.02) | 52.43 (0.14) | 49.37 (0.21) |
| Mixup [61] | ✓ | ✗ | 73.07 (0.08) | 71.52 (0.21) | 69.44 (0.29) | **56.64 (0.02)** | 52.95 (0.06) | 49.97 (0.11) |
| EWC [22] | ✓ | ✗ | **73.64 (0.02)** | 71.53 (0.22) | 68.99 (0.46) | 56.60 (0.04) | 52.78 (0.07) | 49.73 (0.12) |
| SI [60] | ✓ | ✗ | 72.58 (0.07) | 71.50 (0.15) | 69.61 (0.34) | 49.98 (1.09) | 47.27 (2.46) | 44.77 (3.11) |
| A-GEM [10] | ✓ | ✗ | 72.23 (0.06) | 71.16 (0.14) | 69.10 (0.28) | 52.02 (0.99) | 48.91 (2.67) | 46.03 (2.09) |
| DRAIN [3] | ✓ | ✗ | 73.42 (0.02) | 71.75 (0.16) | 69.69 (0.20) | 56.04 (0.06) | 52.07 (0.18) | 48.97 (0.34) |
| **EvoS** | ✓ | ✗ | 73.42 (0.02) | **72.36 (0.05)** | **70.19 (0.08)** | 56.60 (0.02) | **53.15 (0.04)** | **50.19 (0.10)** |

For Huffpost and Arxiv, backbone DistilBERT-base is too big to apply the full GI and DRAIN. So we apply DRAIN only to the classifier and apply GI without the fine-tuning stage.

# E Experimental Results with Error Bars

In this section, we report the mean and standard derivation (denoted as mean (std) ) for each task, when running with 3 random trials. The results with error bars on Yearbook, RMNIST and fMoW are provided in Table 5, and the error bars on Huffpost and Arxiv are given in Table 6.

# F Complexity Analysis

**Time Complexity of MSAM.** Taking one of the multi-head self-attention module $\mathcal{A}_w$ in MSAM as an example, we denote $d_f$ as the feature dimension of its input tokens, and $d_h$, $n_h$ and $n_i$ denote the feature dimension of its heads, the number of its heads and the number of its input tokens, respectively. Then the time complexity of $\mathcal{A}_w$ is $\mathcal{O}((n_i^2 + n_i \cdot d_f) \cdot (d_h \cdot n_h))$. Since $n_i$ will be no larger than the number of training domains $T$ and $d_h \cdot n_h$ is usually set to $d_f$ in transformers, the time complexity of MSAM can be roughly approximated as $\mathcal{O}(W \cdot (T^2 d_f + T \cdot d_f^2))$, where $W$ is the number of multi-head self-attention modules in MSAM. It is roughly equivalent to the time complexity of a single multi-head self-attention layer in conventional transformers multiplied by $W$.

**Memory Complexity of $\mathcal{M}$.** Assuming that there are $T$ historical domains and the dimension of statistic vectors is $d_f$, then the memory complexity of the memory pool $\mathcal{M}$ is $\mathcal{O}(T \cdot d_f)$. In practice, after being processed by deep neural networks, the dimension of pooled features is usually much smaller than that of original inputs. Moreover, only two vectors need to be stored per domain. Hence, the memory cost of $\mathcal{M}$ is relatively small, compared with sample replay-based CL methods. Concretely, Table 7 provides the memory cost of $\mathcal{M}$ and the increment of GPU memory cost for EvoS on Yearbook, RMNIST and fMoW datasets with batch size 64.

Table 7: Memory cost on Yearbook, RMNIST and fMoW datasets.

| GPU memory cost (GB) of different methods | | | |
|---|---|---|---|
| Method | Dataset | | |
| | Yearbook | RMNIST | fMoW |
| | Backbone | | |
| | 4-layer CNN [56] | ConvNet [44] | DenseNet-121 [18] |
| IncFinetune | 1.72 | 1.84 | 10.69 |
| EvoS | 1.94 | 2.09 | 11.04 |
| Increment $\Delta$ (GB) | 0.22 | 0.25 | 0.35 |
| Memory cost (MB) of the memory pool $\mathcal{M}$ | | | |
| EvoS | 5.25 | 9.00 | 32.00 |

Table 8: Model parameters (MB) of different methods on Yearbook, RMNIST and fMoW.

| Method | Dataset | | |
|---|---|---|---|
| | Yearbook | RMNIST | fMoW |
| | Backbone | | |
| | 4-layer CNN [56] | ConvNet [44] | DenseNet-121 [18] |
| LSSAE [44] | 4.70 | 23.25 | 90.92 |
| DRAIN [3] | 7.51 | 184.29 | 1113.85 |
| EvoS | 1.94 | 15.81 | 56.11 |

**Model Complexity.** Here, we measure the model complexity by the number of parameters. Specifically, Table 8 presents the model complexity of our method EvoS and other two temporal DG methods LSSAE [44] and DRAIN [3] on Yearbook, RMNIST and fMoW datasets. LSSAE [44] introduces sequential autoencoder to explore the underlying continuous structure in the latent space of deep neural networks, where the complicated VAE and LSTM networks require lots of parameters. And DRAIN [3] needs to encode and decode the entire network parameters, which also requires a great number of parameters for large backbone networks. By contrast, our EvoS is overall less complex than these temporal DG methods, and is more friendly to the relatively large backbone network.

# G    Additional Results

**Results under Eval-Stream Manner.** In this part, we additionally provide the results when adopting the evaluation manner of Eval-Stream in [56]. Specifically, Eval-Stream denotes the evaluation with domain stream, i.e., the model is evaluated at each timestamp using the average and worst performance on the next $K$ timestamps. Formally, given a sequence of domains with total length $\mathcal{T}$, the average performance $Avg_{stream}$ and worst performance $Worst_{stream}$ under the strategy of Eval-Stream are defined as $Avg_{stream} = \frac{1}{\mathcal{T}-K}\sum_{i=1}^{\mathcal{T}-K}\frac{1}{K}\sum_{j=i+1}^{i+K}Acc_i(\mathcal{D}^j)$, $Worst_{stream} = \frac{1}{\mathcal{T}-K}\sum_{i=1}^{\mathcal{T}-K}\min_{j\in\{i+1,\cdots,i+K\}}Acc_i(\mathcal{D}^j)$, where $Acc_i(\mathcal{D}^j)$ is the accuracy on domain $\mathcal{D}^j$ when using the model at the $i$-th timestamp. And Table 9 shows the results on Yearbook and Huffpost datasets when using the Eval-Stream evaluation manner. According to the results, we see that EvoS still outperforms other baselines, showing that our method is better at handling the problem of continual domain generalization over temporal drift.

Table 9: Accuracy (%) on Yearbook and Huffpost under the evaluation manner of **Eval-Stream**. (Yearbook: $\mathcal{T} = 21$, $K = 5$, Huffpost: $\mathcal{T} = 7$, $K = 3$)

| Method | Conference | Yearbook Accuracy (%) ↑ | | Huffpost Accuracy (%) ↑ | |
|---|---|---|---|---|---|
| | | $Avg_{stream}$ | $Worst_{stream}$ | $Avg_{stream}$ | $Worst_{stream}$ |
| IncFinetune | - | 89.67 | 83.56 | 67.97 | 64.11 |
| Mixup [61] | ICLR'18 | 84.79 | 78.69 | 67.33 | 63.48 |
| SimCLR [11] | ICML'20 | 89.50 | 83.16 | - | - |
| SwAV [8] | NeurIPS'20 | 90.05 | 84.08 | - | - |
| EWC [22] | arXiv'16 | 90.15 | 83.75 | 68.31 | 64.58 |
| SI [60] | ICML'17 | 90.14 | 84.07 | 67.93 | 64.05 |
| A-GEM [10] | ICLR'19 | 84.40 | 77.59 | 65.83 | 62.20 |
| DRAIN [3] | ICLR'23 | 87.26 | 81.95 | 68.04 | 64.14 |
| **EvoS** | - | **90.43** | **84.48** | **68.66** | **64.97** |

Table 10: Misclassification error (%) on 2-Moons, ONP and Elec2. ($K = 1$)

| Method | Conference | Misclassification error (in %) ↓ | | |
|--------|-----------|---------|-----|-------|
| | | 2-Moons | ONP | Elec2 |
| Offline | - | 22.4±4.6 | 33.8±0.6 | 23.0±3.1 |
| LastDomain | - | 14.9±0.9 | 36.0±0.2 | 25.8±0.6 |
| IncFinetune | - | 16.7±3.4 | 34.0±0.3 | 27.3±4.2 |
| CDOT [43] | arXiv'19 | 9.3±1.0 | 34.1±0.0 | 17.8±0.6 |
| CIDA [55] | ICML'20 | 10.8±1.6 | 34.7±0.6 | 14.1±0.2 |
| GI [41] | NeurIPS'21 | 3.5±1.4 | 36.4±0.8 | 16.9±0.7 |
| DRAIN [3] | ICLR'23 | 3.2±1.2 | 38.3±1.2 | 12.7±0.8 |
| **EvoS** | - | **2.5±1.0** | **33.1±0.6** | **11.6±0.7** |

**Results on More Datasets.** In addition to the datasets provided in [56], we also run our method on the *2-Moons*, *ONP* and *Elec2* datasets used in [3]. Specifically, *2-Moons* is a variant of the 2-entangled moons dataset by rotating data counter-clockwise in units of $18°$ to construct 10 domains, where the rotation angle is used to simulate the temporal shift. For 2-Moons, we use Adam optimizer and a MLP with two hidden layers of hidden size 64 and 128, and $B = 64, lr = 1e-3, \alpha = 1.0, \lambda = 1.0, W = 3, d_h = 8, n_h = 32, T = 9, K = 1$. ***Online News Popularity (ONP)*** summarizes a heterogeneous set of features about articles published by Mashable in a period of two years. The dataset is split into 6 domains by time and the goal is to predict the number of shares in social networks (popularity). For ONP, we use Adam optimizer and a MLP with one hidden layer of hidden size 128, and $B = 64, lr = 1e-4, \alpha = 1.0, \lambda = 1.0, W = 3, d_h = 8, n_h = 32, T = 5, K = 1$. ***Electrical Demand (Elec2)*** contains information about the demand of electricity in a particular province. Following [3, 41], the first 30 domains in Elec2 are used (two weeks as one time domain) and the task is to predict whether the demand of electricity in each period (of 30 mins) is higher or lower than the average demand over the last day. For Elec2, we use Adam optimizer and a MLP with two hidden layers of hidden size 128 and 128, and $B = 64, lr = 1e-4, \alpha = 1.0, \lambda = 1.0, W = 3, d_h = 8, n_h = 32, T = 29, K = 1$. For these datasets, as in [3, 41], we use the last domain for testing and the rest for training. Please refer to [3] for more dataset details. The experimental results are given in Table 10, where the misclassification errors of compared baselines are all reported from DRAIN [3]. From Table 10, we can observe that our method EvoS still surpasses the most recent method DRAIN, affirming its effectiveness in temporal domain generalization.

Table 11: Data Subset Size for the Yearbook Dataset.

| Domain | Interval | Training Split | Validation Split | All |
|--------|----------|----------------|------------------|-----|
| 1 | 1930 - 1933 | 758 | 87 | 845 |
| 2 | 1934 - 1937 | 1149 | 130 | 1279 |
| 3 | 1938 - 1941 | 949 | 108 | 1057 |
| 4 | 1942 - 1945 | 2353 | 263 | 2616 |
| 5 | 1946 - 1949 | 1229 | 138 | 1367 |
| 6 | 1950 - 1953 | 1082 | 122 | 1204 |
| 7 | 1954 - 1957 | 1646 | 185 | 1831 |
| 8 | 1958 - 1961 | 1295 | 146 | 1441 |
| 9 | 1962 - 1965 | 1468 | 166 | 1634 |
| 10 | 1966 - 1969 | 2227 | 249 | 2476 |
| 11 | 1970 - 1973 | 1634 | 183 | 1817 |
| 12 | 1974 - 1977 | 2238 | 250 | 2488 |
| 13 | 1978 - 1981 | 1553 | 175 | 1728 |
| 14 | 1982 - 1985 | 2331 | 261 | 2592 |
| 15 | 1986 - 1989 | 1792 | 201 | 1993 |
| 16 | 1990 - 1993 | 1729 | 195 | 1924 |
| 17 | 1994 - 1997 | 1882 | 211 | 2093 |
| 18 | 1998 - 2001 | 2136 | 239 | 2375 |
| 19 | 2002 - 2005 | 1868 | 210 | 2078 |
| 20 | 2006 - 2009 | 1010 | 114 | 1124 |
| 21 | 2010 - 2013 | 1102 | 125 | 1227 |
| total | 1930 - 2013 | 33431 | 3758 | 37189 |

Table 12: Data Subset Size for the fMoW Dataset.

| Domain | Year | Training Split | Validation Split | All |
|---|---|---|---|---|
| 1 | 2002 | 1676 | 227 | 1903 |
| 2 | 2003 | 2279 | 276 | 2555 |
| 3 | 2004 | 1755 | 240 | 1995 |
| 4 | 2005 | 2512 | 324 | 2836 |
| 5 | 2006 | 3155 | 406 | 3561 |
| 6 | 2007 | 1497 | 190 | 1687 |
| 7 | 2008 | 2261 | 298 | 2559 |
| 8 | 2009 | 7439 | 935 | 8374 |
| 9 | 2010 | 18957 | 2456 | 21413 |
| 10 | 2011 | 22111 | 2837 | 24948 |
| 11 | 2012 | 24704 | 3138 | 27842 |
| 12 | 2013 | 3465 | 385 | 3850 |
| 13 | 2014 | 5572 | 620 | 6192 |
| 14 | 2015 | 8885 | 988 | 9873 |
| 15 | 2016 | 14363 | 1596 | 15959 |
| 16 | 2017 | 5534 | 615 | 6149 |
| total | 2002-2017 | 126165 | 15531 | 141696 |

Table 13: Data Subset Size for the Huffpost Dataset.

| Domain | Year | Training Split | Validation Split | All |
|---|---|---|---|---|
| 1 | 2012 | 6701 | 744 | 7446 |
| 2 | 2013 | 7492 | 832 | 8325 |
| 3 | 2014 | 9539 | 1059 | 10599 |
| 4 | 2015 | 11826 | 1313 | 13140 |
| 5 | 2016 | 10548 | 1172 | 11721 |
| 6 | 2017 | 7907 | 878 | 8786 |
| 7 | 2018 | 3501 | 388 | 3890 |
| total | 2012-2018 | 57514 | 6386 | 63907 |

Table 14: Data Subset Size for the Arxiv Dataset.

| Domain | Year | Training Split | Validation Split | All |
|---|---|---|---|---|
| 1 | 2007 | 131550 | 14616 | 146167 |
| 2 | 2008 | 62460 | 6939 | 69400 |
| 3 | 2009 | 206244 | 22916 | 229161 |
| 4 | 2010 | 50665 | 5629 | 56295 |
| 5 | 2011 | 55741 | 6193 | 61935 |
| 6 | 2012 | 51678 | 5741 | 57420 |
| 7 | 2013 | 64951 | 7216 | 72168 |
| 8 | 2014 | 79498 | 8833 | 88332 |
| 9 | 2015 | 193979 | 21553 | 215533 |
| 10 | 2016 | 120682 | 13409 | 134092 |
| 11 | 2017 | 111024 | 12336 | 123361 |
| 12 | 2018 | 123891 | 13765 | 137657 |
| 13 | 2019 | 142767 | 15862 | 158630 |
| 14 | 2020 | 166014 | 18445 | 184460 |
| 15 | 2021 | 201241 | 22360 | 223602 |
| 16 | 2022 | 89765 | 9973 | 99739 |
| total | 2007-2022 | 1852150 | 205786 | 2057952 |

