# OpenReview forum: "Evolving Standardization for Continual Domain Generalization over Temporal Drift"
_NeurIPS.cc/2023/Conference — NeurIPS 2023 poster_

### Official Review · Reviewer_YTjU · 2023-06-26

**Soundness:** 3 good
**Presentation:** 4 excellent
**Contribution:** 4 excellent
**Rating:** 6
**Confidence:** 3

**Summary:**

Mitigating the temporal distribution shift problem is meaningful in realistic. This paper introduces the problem of continual domain generalization over temporal drift (CDGTD), aiming to address the issue of potential temporal distribution shifts in unseen test domains. The authors propose an Evolving Standardization method that utilizes a multi-scale attention module to forecast the distribution of the unseen test domain. They then mitigate the distribution shift by standardizing features using the generated statistics of the test domain. To the best of my knowledge, the method is novel and shows promising results in the experiments section.

**Strengths:**

1. The paper exhibits a commendable structure and offers clear explanations throughout.
2. The authors conduct ablation studies to examine the individual components of their proposed method.
3. To the best of my knowledge, the method presented in this paper is novel.
4. Experiments results confirm the proposed method works well.

**Weaknesses:**

1. The authors only selected a subset of datasets from the Wilds-Time benchmark for their experiments, it would be valuable to include results on MIMIC-Readmission and MIMIC-Mortality datasets as well, considering wilds-time is the most relevant benchmark for this problem.

**Questions:**

1. Figure 3 (b) shows that the method is not sensitive to hyperparameters \alpha and \lambda on yearbook datasets which is pretty strange to me. Do the authors have some intuitions about that? If I am a user using your method, Could the authors provide suggestions or considerations for selecting appropriate values for these hyperparameters? In particular, I am curious about the rationale behind the specific choices of {0.1, 0.5, 1.0, 1.5, 2.0} for the ablation experiment.

2. What is the difference between your experimental settings and wilds-time's eval-stream? Why don't directly follow wilds-time for the experiments?


**Limitations:**

NIL

---

> ### Author Rebuttal · Authors · 2023-08-10
>
> Sincerely thanks for your efforts in reviewing the paper. Below, we respond to your questions in detail.
>
> > **Q1:** The used datasets are only a subset of datasets from the Wilds-Time benchmark. It would be valuable to include MIMIC-Readmission and MIMIC-Mortality datasets.
>
> **A1:** Thanks for your suggestion. Actually, we almost use all the provided datasets in the Wilds-Time benchmark, except for the MIMIC-Readmission and MIMIC-Mortality datasets. This omission is not deliberate. The reason is that they are restricted-access resources and we have some troubles in acquiring the credentialed access to the two medical datasets. Since Yearbook, fMoW, Huffpost and Arxiv are all from the Wilds-Time benchmark and our method has achieved comparable and even superior performance on the four datasets, we think these results are sufficient to validate the effectiveness of our method. Yet, of course, there is no doubt that more results are more convincing, and we will leave the successful application for the credentialed access and the results for the two medical datasets as future work.
>
> As a compensation for the missing results of these two datasets, we additionally run our method on some other datasets (2-moons, Online News Popularity (ONP) and Electrical Demand (Elec2)) that are used by previous temporal DG methods GI [2] and DRAIN [3]. These datasets are also of distribution shifts over time. Please refer to the global reply **R1** for the detailed dataset description. **Table 4** in the PDF file gives the results, where the misclassification errors of baselines are reported from DRAIN [3]. We see that our method EvoS still outperforms the most recent method DRAIN.
>
> > **Q2:** Explanation for the insignificant hyperparameter sensitivity and suggestion for hyperparameter values.
>
> **A2:** Thanks. The hyperparameters $\alpha$ and $\lambda$ control sampling truncation range and the tradeoff of adversarial loss $\mathcal{L}\_{adv}$, respectively. For the normal distribution, values within one standard deviation from the mean make up 68.27%, two deviations account for 95.45%, and three deviations cover 99.73%. Since $\alpha=3$ is akin to no truncation (Variant G in the ablation study of the paper), we limit $\alpha$ to 2 in the sensitivity experiment, using an interval of 0.5 for other values. For $\lambda$, as $\mathcal{L}\_{adv}$ is of similar magnitude as cross-entropy loss $\mathcal{L}\_{ce}$, we vary $\lambda$ by 0.5 intervals in both directions around $\lambda=1.0$.
>
> Actually, in the original Figure 3(b) of the paper, sensitive regions exist (e.g., $\lambda \in [1.0, 2.0]$). However, due to the narrow value range, sensitivity might not be evident. In **Figure 2(b)** of the PDF file, we extend hyperparameter ranges to $\alpha \in \\{0.1, 0.5, 1.0, 1.5, 2.0, 2.5, 3.0\\}$ and $\lambda \in \\{0.1, 0.5, 1.0, 1.5, 2.0, 5.0, 10.0, 15.0, 20.0\\}$. These new results exhibit similar conclusions that EvoS is more sensitive to $\lambda$ than $\alpha$, and large $\lambda$ values significantly worsen performance. Thus, in practice, we recommend selecting $\lambda$ from a range $\leq$1 through validation. For $\alpha$, its effect is modest; users can directly use $\alpha=1$, which balances diversity and representativeness, as about 68.27% of the data lie within one standard deviation from the mean in a normal distribution.
>
> > **Q3:** Difference between our experimental setting and wilds-time's eval-stream.
>
> **A3:** Thanks. Wild-Time [1] provides two evaluation strategies: Eval-Fix and Eval-Stream, and our experimental setting uses the Eval-Fix. Eval-Fix denotes that the model is evaluated on a single, fixed train-test time split. This evaluation strategy offers a quick evaluation protocol and is the primary evaluation strategy in Wild-Time. Thus, we adopt the Eval-Fix strategy in our experiments.
> Specifically, given a sequence of domains with total length $\mathcal{T}$ and the split timestamp $T\_s$, the average performance $Avg\_{fix}$ (i.e., "OOD avg." in our paper) and worst performance $Worst\_{fix}$ (i.e., "OOD worst" in our paper) under the strategy of Eval-Fix are defined as
>
> $Avg\_{fix} = \frac{1}{K} \sum_{i=T\_s+1}^{T\_s+K} {Acc(\mathcal{D}^i)}, Worst\_{fix} = \min\_{i\in \\{T\_s+1, \cdots, T\_s+K\\}} {Acc(\mathcal{D}^i)},$
>
> where $K=\mathcal{T} - T\_s$ is the number of future domains to be evaluated, and $Acc(\mathcal{D}^i)$ is the accuracy on domain $\mathcal{D}^i$.
>
> As for Eval-Stream, it denotes the evaluation with data stream, i.e., the model is evaluated at each timestamp using the average and worst performance on the next $K$ timestamps. Concretely, given a sequence of domains with total length $\mathcal{T}$, the average performance $Avg\_{stream}$ and worst performance $Worst\_{stream}$ under the strategy of Eval-Stream are defined as
>
> $Avg\_{stream} = \frac{1}{\mathcal{T}-K} \sum_{i=1}^{\mathcal{T}-K} \frac{1}{K} \sum_{j=i+1}^{i+K} Acc_{i}(\mathcal{D^j}), Worst\_{stream} = \frac{1}{\mathcal{T}-K} \sum_{i=1}^{\mathcal{T}-K} \min_{j\in \\{i+1, \cdots, i+K\\}} Acc_{i}(\mathcal{D^j}),$
>
> where $Acc_{i}(\mathcal{D^j})$ is the accuracy on domain $\mathcal{D^j}$ when using the model at the $i$-th timestamp. Hence, the Eval-Fix that we adopt can be viewed as a single timestamp evaluation within Eval-Stream, where the model is only evaluated at $T\_s$ and $K=\mathcal{T}-T\_s$.
>
> To further verify the effectiveness of our method, in **Table 3** of the PDF file, we additionally provide the result when using the Eval-Stream evaluation strategy. Due to time limitations, we just provide results on Yearbook and Huffpost datasets. According to the result, we see that EvoS still outperforms other baselines, showing that our method is better at handing the problem of continual domain generalization over temporal drift.

---

> > ### Comment · Reviewer_YTjU · 2023-08-16
> >
> > Thank you for the clarifying comments. I will keep my score. I suggest include MIMIC-Readmission and MIMIC-Mortality datasets in your future version to ensure a comprehensive evaluations.

---

### Official Review · Reviewer_uaS8 · 2023-06-27

**Soundness:** 3 good
**Presentation:** 3 good
**Contribution:** 3 good
**Rating:** 6
**Confidence:** 4

**Summary:**

This paper introduces a problem formulation for Continual Domain Generalization over Temporal Drift (CDGTD) and proposes the Evolving Standardization (EvoS) method to address the challenge of gradually shifting data distributions over time, aiming to generalize to unseen domains that are not too far into the future. The EvoS method characterizes the evolving pattern of feature distribution and mitigates distribution shift by standardizing features with generated statistics of the corresponding domain. It utilizes a multi-scale attention module (MSAM) to learn the evolving pattern under sliding time windows of varying lengths. MSAM can also generate statistics of the current domain based on the previous domains' statistics and the learned evolving pattern. The paper demonstrates the efficacy of EvoS through experiments on multiple real-world datasets, including images and texts.

**Strengths:**

1. The paper aims to addresses an important problem in machine learning, namely, Continual Domain Generalization over Temporal Drift. The proposed algorithm, EvoS, is a new approach which characterizes the evolving pattern of feature distribution and mitigates the distribution shift by standardizing features with generated statistics of the corresponding domain.
2. The paper highlights the necessity of learning evolving patterns to address temporal distribution shift.
3. The paper conducts experiments on multiple real-world datasets, including images and texts, to validate the efficacy of the proposed EvoS method.

**Weaknesses:**

1. The paper presents a setting where domains arrive sequentially, and models can only access the data of the current domain. However, the authors argue that we should learn from the current source domain to adjust models and generate target domains. This setting contradicts the idea of Domain Generalization, where data from multiple domains may be available simultaneously, or models can access historical data from previous domains.

2. However, the authors of the paper do not provide sufficient context or motivation for the research problem and fail to distinguish this setting from continuous learning and test-time adaptation.

3. The EvoS method introduced in this paper is notably more complex than other baselines. However, the authors do not provide a detailed analysis of the computational requirements of the proposed method.

4. EvoS outperforms other baselines in its tailor-made tasks, which make the efficacy of EvoS unconvincing.

**Questions:**

Please refer to the weakness section.

**Limitations:**

Please refer to the weakness section.

---

> ### Author Rebuttal · Authors · 2023-08-10
>
> Thanks for your efforts in reviewing the paper. Below, we address your concerns in detail.
>
> > **Q1:** This setting contradicts the idea of Domain Generalization.
>
> **A1:** Thanks. Firstly, we want to clarify that our CDGTD is a challenging and practical variant of conventional DG. Hence, its setting naturally differs.
>
> Past DG methods focus on static, discrete scenarios with fixed domains and sudden shifts among them. They train a model on multiple domains simultaneously offline, striving for ideal but hardly achievable generalization on any unsee domain. In real world, new data continuously arises over time. A realistic way is to leverage fresh domain along with old domains to enhance model generalization. One way is to store past domains and re-train the model using standard DG methods. Yet, this is inefficient and doesn't allow reuse of previous model. Besides, retaining all historical domains requires substantial memory, especially in scenarios with rapid data accumulation.
>
> Considering the limitations, we draw inspiration from continual learning (CL) to consider DG in a dynamic and continuous configuration. In CDGTD, the training domain evolves dynamically over time following some patterns, and domains are assumed to arrive sequentially to mimic the realistic scenario where new training domain emerges, while the access to only current domain minimizes storage cost. Like CL, the model starts from the previous timestamp's state and incrementally trains with current domain. This makes previously trained model reusable and boosts training efficiency.
>
> Overall, CDGTD aims at a more practical DG scenario that simultaneously considers the **dynamics of training domains**, **training efficiency** for frequently updating model for better generalization and **storage burden**.
>
> > **Q2:** Motivation for CDGTD and differences with continuous learning (CL) and test-time adaptation (TTA).
>
> **A2:** Thanks. Actually, in the introduction, we have explained why CDGTD is introduced. For your convenience, we summarize it below.
>
> **Motivation of CDGTD.** In standard DG, models are trained offline on fixed source domains to achieve broad generalization on any unseen domain. However, such ideal generalization is tough. In real world, new data continually emerges over time, offering a chance to enhance the generalization of previously trained model. But standard DG methods struggle to update models with fresh data efficiently. Their static domain setup and offline training mode rquire starting model training anew with new and prior data, leading to low efficiency and high storage cost for historical data. Instead, a more practical way is to employ the training paradigm of CL, but with the goal to generalize on future unseen domains. This way reduces storage cost and enhances training efficiency. Thus, we introduce CDGTD to efficiently and effectively address temporal DG.
>
> **Differences with CL.** The biggest difference is the objective. CL aims to learn new tasks while retaining performance on old tasks, whereas CDGTD prioritizes generalization on future unseen domains. These different goals yield distinct challenges. CL faces catastrophic forgetting of task-specific knowledge, while CDGTD tackles modeling underlying evolutionary patterns of temporal domains, and how to utilize these patterns to mitigate distribution shifts in forthcoming domains.
>
> **Differences with TTA.**  One difference is the focused optimization phase. TTA optimizes the source-pretrained model during testing phase with test data, while CDGTD optimizes the model during training phase with sequential training domains. Another difference is the data. TTA's arriving test data usually comes from the same domain or distribution, while CDGTD's test domain evolves over time, showing distribution shifts. Moreover, TTA emphasizes in-time adaptation using test data, while CDGTD focuses on generalization without using test data.
>
> > **Q3:** Analysis of the computational requirements of the method.
>
> **A3:** Thanks. Please refer to the global reply **R2** for the detailed analysis of memory and time complexity.
>
> > **Q4:** EvoS outperforms other baselines in its tailor-made tasks, which make the efficacy of EvoS unconvincing.
>
> **A4:** Firstly, the challenging CDGTD remains mostly unexplored. It is inevitable to run related methods in our experimental setup for comparisons. To ensure fairness, we have applied the same training and evaluation settings for compared methods.
>
> Secondly, as described in Section 4.1, our experimental tasks mainly comes from the existing Wild-Time benchmark (NeurIPS'22) [1]. This benchmark offers multiple **real-world** datasets displaying distribution shifts over time, including the Yearbook, fMoW, Huffpost, and Arxiv datasets we use. By contrast, prior temporal DG methods GI [2] and DRAIN [3] use either synthetic (e.g., 2-Moons) or small (e.g., Shuttle) datasets. Thus, we opt to assess performance on more challenging and realistic datasets.
>
> Thirdly, the evaluation strategy is not specially tailor-made by us. It is from the Eval-Fix strategy in Wild-Time [1]. Specifically, given a domain sequence of length $\mathcal T$ and a split timestamp $T\_s$, the model trains on $\mathcal{D}^1$ to $\mathcal{D}^{T_s}$ sequentially and evaluates on future domains $\mathcal{D}^{T\_s+1}$ to $\mathcal{D}^{\mathcal T}$. Under Eval-Fix, average performance $OOD_{avg}$ and worst performance $OOD_{worst}$ are defined as
>
> $OOD_{avg} = \frac{1}{\mathcal T - T\_s} \sum_{i=T\_s+1}^{\mathcal T} {Acc(\mathcal{D}^i)}, OOD_{Worst} = \min_{i\in \\{T_s+1, \cdots, \mathcal T\\}} {Acc(\mathcal{D}^i)}.$
>
> Finally, to address your concerns about convincingness, we present the generalization results on the last domain for 2-Moons, ONP, and Elec2 datasets from DRAIN [3] in **Table 4** of the PDF file. Notably, all baselines' results are reported from DRAIN. The outcomes show EvoS's superior performance on these non-tailored tasks, affirming its effectiveness.

---

> > ### Comment · Reviewer_uaS8 · 2023-08-14
> >
> > Thank you for the authors' responses. I appreciate the clarifications provided regarding the proposed settings in this paper. I find the setting to be meaningful. However, I still have concerns regarding the complexity of the model. Therefore, I will incrementally raise my rating. Nonetheless, I remain open to reassessing and further improving my rating if the other reviewers and ACs consider these concerns to be of lesser significance.

---

> > > ### Author Response · Authors · 2023-08-15
> > > **Further response to concerns about the complexity of the model**
> > >
> > > We sincerely thank you for recognizing our clarifications of the proposed setting. For your concerns about the complexity of the model, we would like to provide a further clarification. In fact, our model is not complex in implementation, where there are only three modules: multi-scale attention module, feature standardization module and adversarial learning module, along with a basic backbone network (e.g., DenseNet-121 for fMoW) for feature extraction.
> > > * For the multi-scale attention module (MSAM), each $\mathcal{A}_w$ has the similar structure with a single multi-head self-attention (MHSA) layer in transformer models. The only special handling is the processing of input tokens, where a sliding time window of length $w$ is applied to input tokens for aggregating information at scale $w$.
> > >
> > > * For the feature standardization module, it just leverage the generated statistics (mean and variance) by MSAM to mitigate the distribution shift by normalization operation.
> > >
> > > * And the adversarial learning module is similar to common ways in DA/DG works, but features are sampled from the preserved domain distributions in the memory pool $\mathcal{M}$ to solve the issue of unavailable historical data.
> > >
> > > Overall, each module has its own role and is not complicated to implement. In addition, we detailedly analyze from the model, time and memory complexities below.
> > >
> > > **Model complexity.** Below, we provide the model complexity (measured by the number of parameters) of our method EvoS and other two temporal DG methods (LSSAE [2] and DRAIN [3]) on the three image classification datasets: Yearbook, RMNIST and fMoW.
> > >
> > > -----------------Parameters (MB) of different methods----------------
> > > | Method | Yearbook | RMNIST | fMoW |
> > > | --- | --- | --- | --- |
> > > | backbone | 4-layer CNN in [1] | ConvNet in [2]| DenseNet-121 |
> > > | LSSAE [2] | 4.70 | 23.25 | 90.92
> > > | DRAIN [3] | 7.51 | 184.29 | 1113.85 |
> > > | EvoS | **1.94** | **15.81** | **56.11**|
> > >
> > > LSSAE introduces Sequential Autoencoder to explore the underlying continuous structure in the latent space of deep neural networks, where the complicated VAE and LSTM networks require lots of parameters. DRAIN [3] needs to encode and decode the entire network parameters, which requires a great amount of parameters for large backbone networks. By contrast, our EvoS is overall less complex than these temporal DG methods, and is more friendly to the relatively large backbone network.
> > >
> > > **Time complexity.** In global reply **R2**, we provide the time complexity of our MSAM, which is approximately $\mathcal{O}(W \cdot (T^2 d_f + T \cdot d_f^2))$. $W$ is the number of multi-head attention modules in MSAM, $T$ is the number of training domains and $d_f$ is the dimension of deep features. In implementation, $W$ is set to a relatively small value ($W=3$) in our paper. Meanwhile, for conventional transformers, the time complexity of a single MHSA layer is $\mathcal{O}(n^2 d + n \cdot d^2)$, where $n$ is the number of input tokens and $d$ is the feature dimension of input tokens. We can see that the time complexity of our MSAM is equivalent to multiplying the time complexity of a single MHSA layer in conventional transformers by a small value of $W$. In other words, it can be seen as $W$ ($W=3$ in our paper) layers of MHSA, which is a quite tiny module.
> > >
> > > **Memory complexity.** In global reply **R2**, we give the memory complexity of our memory pool $\mathcal{M}$, which is $\mathcal{O}(T\cdot d_f)$. $T$ is the number of training domains and $d_f$ is the feature dimension. In practice, $d_f$ is usually much smaller than the dimensions of original inputs. For example, the dimensions of an image in the fMoW dataset are $224\times 224 \times 3 = 150,528$, while the dimension of pooled features in DenseNet-121 is $1,024$, about $0.007$ times of the former. Besides, only two vectors need to be stored per domain. Hence, the memory cost of $\mathcal{M}$ is relatively small. The following table lists the memory cost of $\mathcal{M}$ and the increment of GPU memory consumption for our EvoS on different datasets with batch size 64.
> > >
> > > ----------Table 1: Memory cost (MB) of the memory pool $\mathcal{M}$ in EvoS ----------
> > > | Method | Yearbook | RMNIST | fMoW |
> > > | --- | --- | --- | --- |
> > > | EvoS | 5.25 | 9.00 | 32.00 |
> > >
> > > ----------Table 2: GPU memory consumption (GB) of different methods----------
> > > | Method | Yearbook | RMNIST | fMoW |
> > > | --- | --- | --- | --- |
> > > | backbone | 4-layer CNN in [1] | ConvNet in [2]| DenseNet-121 |
> > > | IncFinetune | 1.72 | 1.84 | 10.69 |
> > > | EvoS | 1.94 | 2.09 | 11.04 |
> > > | Increment $\Delta$ (GB)| 0.22 | 0.25 | 0.35 |
> > >
> > > From the above three aspects of complexity, we think our method has an acceptable level of complexity.
> > >
> > > Refs:
> > >
> > > [1] Wild-time: A benchmark of in-the-wild distribution shift over time. In NeurIPS, 2022.
> > >
> > > [2] Generalizing to evolving domains with latent structure-aware sequential autoencoder. In ICML, 2022.
> > >
> > > [3] Temporal domain generalization with drift-aware dynamic neural networks. In ICLR, 2023.

---

> > > > ### Author Response · Authors · 2023-08-18
> > > > **Looking forward to seeing your feedback!**
> > > >
> > > > Dear reviewer uaS8,
> > > >
> > > > We have provided a further response to the complexity of our method from three aspects of **model**, **time** and **memory complexity**. We hope the quantitative results of **fewer model parameters** and **tiny increments of GPU memory consumption** has addressed your concerns about the complexity of the model. The rebuttal period is going to end. If you have any other questions or suggestions, please let us know and we are more than happy to respond as soon as possible. Looking forward to your feedback. Thanks so much!
> > > >
> > > > Best regards,
> > > >
> > > > Authors

---

> > > > ### Comment · Reviewer_uaS8 · 2023-08-19
> > > >
> > > > Thank you for your clarifying comments. I will adjust my score accordingly.

---

### Official Review · Reviewer_P1pd · 2023-07-03

**Soundness:** 3 good
**Presentation:** 3 good
**Contribution:** 3 good
**Rating:** 8
**Confidence:** 5

**Summary:**

This paper introduces the problem of continual domain Generalization over Temporal Drift (CDGTD), where the domain distribution gradually changes over time and the model needs to generalize to new domains in the near future with training domains sequentially arriving. And this paper also proposes an Evolving Standardization (EvoS) approach to predict and adapt the data distribution of future domains by utilizing a multi-scale attention module to generate the mean and standard deviation of features in current domain and conducting feature standardization based on generated statistics to mitigate the domain shift. Extensive experiments on both image recognition and text classification are done to validate the effectiveness of the proposed method.

**Strengths:**

1. The introduction of the problem of continual domain generalization over temporal drift (CDGTD) is innovative and meaningful. Unlike existing temporal domain generalization approaches, CDGTD incorporates incremental scenarios that mimic real-world situations where training domains arrive sequentially. This assumption adds an extra challenge to generalization, as the model needs to mitigate catastrophic forgetting during the continual learning process.

2. The proposed method EvoS is novel to me. Especially, the inclusion of multi-scale attention module to generate evolved feature statistics is intriguing, where sliding time windows of different lengths are cleverly used to integrate multiscale information for better characterization of evolving patterns.

3. Overall, the quality of the paper is commendable. The authors have conducted a thorough ablation study to examine the impact of various components, and they have also made the code and datasets available in the supplementary material.

4. The paper is mostly well-organized and well-written.

5. The experimental results on image recognition and text classification datasets in the CDGTD setting demonstrate state-of-the-art generalization performance across a diverse range of tasks.



**Weaknesses:**

1. I am wondering whether the number of heads (n_h) and the feature dimension of heads (d_h) are the same in each attention module A_w, since the authors just simply describe the value of n_h and d_h in MSAM. Yet, there are multiple attention modules in MSAM and the input dimension of each attention module is different. More details are necessary to make it clearer.

2. In Eq. (12) and (13), it seems that the random sampling process is only performed once at the beginning of each training phase. Wouldn't this reduce the diversity and representativeness of the proxy samples for historical domains?

3. As the number of historical domains increases, will the adversarial training suffer from the imbalance problem? What if we randomly select one historical domain in each iteration to participate in the adversarial training?

4. The specific training way of “Offline” and “IncFinetune” in Table 1 and 2 should be provided.


**Questions:**

Minor issue, typos:
Line 299: Finally, the results of variant H and EvoS compares ...
Line 309 and 319: MASM -> MSAM
Line 321: \beta and \lambda controls ...


**Limitations:**

Limitations of the proposed approach have been discussed in the appendix.

---

> ### Author Rebuttal · Authors · 2023-08-10
>
> We are grateful for your efforts in reviewing the paper as well as your constructive comments. Below, we do our utmost to address your concerns.
>
> > **Q1:** I am wondering whether the number of heads ($n\_h$) and the feature dimension of heads ($d\_h$) are the same in each attention module $\mathcal{A}\_w$, since the authors just simply describe the value of $n\_h$ and $d\_h$ in MSAM. Yet, there are multiple attention modules in MSAM and the input dimension of each attention module is different. More details are necessary to make it clearer.
>
> **A1:** Thanks for pointing out this. Actually, in practical implementation, we use the same feature dimension of heads $d\_h$ for each attention module $\mathcal{A}\_w$ in MSAM, $w=1,2,\cdots, W$, while the number of heads in attention module $\mathcal{A}\_w$ is set to $w\cdot n\_h$ to accommodate the different input dimension of each attention module. For the specific values of $d\_h$ and $n\_h$, we have provided in Section D.2 of the appendix ($d\_h=8$ for all datasets and $n\_h=16$ for Yearbook, $n\_h=32$ for RMNIST, $n\_h=64$ for fMoW, $n\_h=128$ for Huffpost and Arxiv). In the revision, we will make this implementation details more clear.
>
> > **Q2:** In Eq. (12) and (13), it seems that the random sampling process is only performed once at the beginning of each training phase. Wouldn't this reduce the diversity and representativeness of the proxy samples for historical domains?
>
> **A2:** Thanks. In fact, the random sampling process is performed in each iteration to provide diverse and plenty features for representing historical domains. Concretely, in each iteration, we will randomly sample a batch of features from every preserved domain distribution in the memory pool $\mathcal{M}$, and use them for the adversarial training to mitigate the overfitting of the model to current domain and enhance the generalizability of the feature encoder. This point will be clarified in the revision.
>
> > **Q3:** As the number of historical domains increases, will the adversarial training suffer from the imbalance problem? What if we randomly select one historical domain in each iteration to participate in the adversarial training?
>
> **A3:** Thanks for your comment. Although the total number of sampled features is greater than the number of features from current domain, the averaging operation in computing the adversarial loss balances the contribution of each feature to the adversarial training, which avoids suffering from the quantity imbalance problem. As for the results when randomly selecting one historical domain distribution in each iteration to participate in the adversarial training, we provide them in **Table 1** of the uploaded PDF file, where one distribution is randomly selected from the memory pool $\mathcal{M}$ in each iteration to sample a batch of $B$ features for calculating the adversarial loss $\mathcal{L}_{adv}$. From the results, we see that randomly selecting one historical domain distribution performs worse. This may be because the feature space to be aligned frequently changes if using this manner, making the optimization challenging. By contrast, it is more stable to simultaneously leverage all the preserved domain distributions in the memory pool $\mathcal{M}$ in each iteration for the adversarial training.
>
> > **Q4:** The specific training way of “Offline” and “IncFinetune” in Table 1 and 2 should be provided.
>
> **A4:** Thanks for your advice. "Offline" denotes merging all training domains into one domain and training the model on the merged domain with the cross-entropy loss for classification tasks. And "IncFinetune" represents incrementally fine-tuning the model, i.e., the model is sequentially trained on each training domain. In other words, at each timestamp, the model is initialized into the state at the previous timestamp and then fine-tuned using the domain at current timestamp. We will add a detailed description of the training for the two baselines in the revised paper.
>
> > **Q5:** Minor issue, typos: Line 299: Finally, the results of variant H and EvoS compares ... Line 309 and 319: MASM -> MSAM Line 321: \beta and \lambda controls ...
>
> **A5:** Thanks. We have proofread the paper carefully and revised the paper thoroughly to correct the typos.

---

> ### Comment · Reviewer_P1pd · 2023-08-13
>
> The rebuttal has addressed my concerns. I believe the proposed setting is inspiring for real-world dynamic environment. As a result, I would like to raise my score to defend my recommendation.

---

### Official Review · Reviewer_Nwuj · 2023-07-05

**Soundness:** 3 good
**Presentation:** 3 good
**Contribution:** 2 fair
**Rating:** 5
**Confidence:** 3

**Summary:**

This paper considers domain generalization over temporal-drift data where the model is trained online and required to generalize to the unseen future domain. The proposed method, Evolving Standardization (EvoS), assumes each domain follows a Gaussian distribution and utilizes transformers to capture the temporal drift by predicting the Gaussian mean and variance of the current domain given historical domains’ means and variances. Specifically, a multi-scale attention module (MSAM) is proposed, which utilizes a sliding window to aggregate domain statistics within a given time scope and makes the transformer predict based on the aggregated domains. The Gaussian statistics generator (namely, feature encoder) is shared over domains and trained in an adversarial way to achieve domain-invariant property. The proposed method is tested on several real-world temporal DG datasets and achieved good performance.

**Strengths:**

1. The proposed problem, which seems like a combination of evolving/temporal domain generalization and online continual learning, is interesting and novel.

2. The proposed method is shown to achieve optimal performance over various existing methods.

3. The paper is in general clearly written and easy to follow.

**Weaknesses:**

1. The technical contribution is incremental. The single-scale attention is a review of transformer models and domain-invariant adversarial loss is a common tool in either DA/DG works. The key idea of the multi-scale attention module, which is a sliding window-based aggregation of multiple domains’ statistics, is relatively straightforward. Probably some theoretical analyses such as the generalization error bound of the proposed method on the future domain can strengthen the overall technical contribution.

2. The usage of the memory pool is similar to episodic memory in continual learning replay-based methods, and I am wondering about the memory complexity/cost, especially considering the case where the number of temporal domains is large. Also, in online continua learning episodic memory is typically assumed fixed and small to guarantee practical interests, and I am curious if a similar assumption is made in this work.

3. The baselines from the continual learning domain used in this paper are not SOTA (the latest one is A-GEM from 2019), and I encourage the author to compare them with more recent SOTA CL methods. In addition, continual domain adaptation is also a closely related area to this paper, and some methods such as CIDA [1] and CDOT [2] are also interesting to explore.

4. I would encourage the author to consider visualizing the trained model in a gradual manner, such as the visualization of the decision boundary on the Rotated 2-Moons dataset in papers of e.g., CIDA, GI, and DRAIN. Such visualization can serve as qualitative analysis and better demonstrate if the method can truly capture the underlying temporal drift of data.

[1] Wang, Hao, Hao He, and Dina Katabi. "Continuously indexed domain adaptation." Proceedings of the 37th International Conference on Machine Learning. 2020.

[2] Jimenez, G. O., Gheche, M. E., Simou, E., Maretic, H. P., and Frossard, P. Cdot: Continuous domain adaptation using optimal transport. arXiv preprint arXiv:1909.11448, 2019.

**Questions:**

Please refer to my “Weaknesses” section for the questions.

My final score will largely depend on the rebuttal and discussion with other reviewers. I am willing to increase my score if my questions are adequately addressed.

**Limitations:**

As far as I have checked, I did not find potential negative societal impacts in this work. I have listed my concerns in the "Weaknesses" section.

---

> ### Author Rebuttal · Authors · 2023-08-10
>
> Thanks a lot for your efforts in reviewing the paper. Below, we respond to your questions in detail.
>
> > **Q1:** The technical contribution is incremental.
>
> **A1:** Thanks. Firstly, although transformers can model sequential relationships, they require inputs themselves to be of sequentiality, like sentence words in NLP or tailored image patches in CV for ViT. In our case, sequentiality spans domains, not within a domain. Directly using transformers on the current domain data fails to capture evolving temporal patterns. Our efficient and novel solution is to model domain distribution using deep feature stats (mean, variance). We then use these statistics from seen domains as transformer inputs to generate next domain stats. Such way captures evolving patterns and meanwhile avoids heavy storage burden for historical data.
>
> Secondly, we further design the simple yet effective Multi-Scale Attention Module (MSAM) to accommodate and integrate the evolving pattern with different length of observation intervals, which has not been considered by previous temporal DG methods GI [2] and DRAIN [3]. Though the implementation of MSAM is relatively straightforward, the results of variant H *vs.* EvoS in the ablation study of the paper show that MSAM is indeed beneficial.
>
> Thirdly, while domain-invariant adversarial loss is prevalent in DA/DG works, its application is challenging in our context due to inaccessible historical domains. We cleverly address this by modeling domain distribution at the deep feature level and utilizing truncated-sampled features as proxies of historical domains. Moreover, we further alleviate distribution shift by standardizing features via predicted evolving statistics. This approach leverages temporally evolutionary pattern, an aspect overlooked by prior DA/DG methods.
>
> Lastly, beyond our method, our another contribution is the introduced problem of Continual Domain Generalization over Temporal Drift (CDGTD). This is a more challenging and practical variant of DG, considering the **dynamic nature of training domains**, the **efficiency of frequent model updates** with newly collected domain data, and the **storage burden** for historical data.
>
> > **Q2:** The memory complexity/cost and memory pool size.
>
> **A2:** Thanks for your comment. Please refer to the global reply **R2** for the detailed memory complexity/cost. As for the memory pool size, since the dataset used in our paper has a moderate number of domains and the memory cost is small, we do not restrict the size of the memory pool $\mathcal{M}$ in our experiments. Yet, as you concerned, a fixed memory pool size is practical when considering a lifelong process, i.e., $T \rightarrow \infty$. Thus, we additionally conduct the experiment, where the memory pool $\mathcal{M}$ is implemented as a FIFO queue with different fixed size $L$. That is, only the statistics for up to the $L$ most recent historical domains can be stored. The results on Yearbook dataset in **Figure 2(a)** of the PDF file demonstrate that our method generally performs well and a relatively large memory pool size is better for temporal domain generalization. So in practice, we recommend that the memory pool size be as large as possible under affordable memory cost. In the revision, we will clarify this.
>
> > **Q3:** Results for more recent SOTA CL method and continual DA methods CIDA and CDOT.
>
> **A3:** Thanks for your advice. We have added the results of CIDA, CDOT and a more advanced CL method SGP (AAAI'23) [4] on dataset Yearbook and RMNIST in **Table 2** of the PDF file. From the results, we can observe that the three methods obtain inferior performance on future domains.
>
> For CIDA, it simultaneously takes the sample and time (i.e., [x, t]) as the input and aims to learn time-invariant feature representations via a two-party adversarial game. In this method, the temporal distribution shift is assumed to be characterized by the value of time, discarding the other features's dependency on time and dependency on other confounding factors. As a result, it fails to accurately capture the complex temporal drift.
>
> For CDOT, it applies the regularized optimal transport to transport the most recent labeled samples to the future using a learned coupling from past data. Its performance depends on how closely the transported images resemble the true target image, showing limited ability in addressing the persistent temporal drift.
>
> And for SGP [4], it combines orthogonal gradient projections with scaled gradient steps along significant gradient spaces from prior tasks to enhance new learning and minimize forgetting. Despite the same mode of sequential training, our goal differs from SGP.  Our EvoS focuses on generalizing to upcoming unseen domains, whereas SGP aims to excel in both past and present tasks. Consequently, lacking specific temporal drift handling, SGP exhibits lower performance on future domains.
>
> By contrast, our EvoS shows superior performance, owing to the better modeling of the evolutionary pattern in temporal domains via MSAM and the mitigation of the distribution shift by standardizing features with generated statistics.
>
> > **Q4:** Visualization of the decision boundary on Rotated 2-Moons dataset.
>
> **A4:** Good suggestion. **Figure 1** of the PDF file gives the result. The 2-Moons dataset in GI [2] is used, where each moon consists of 100 instances, and 10 domains (0 to 9) are obtained by sampling 200 data points from the 2-Moons distribution and rotating them counter-clockwise in units of $18^\circ$. In Figure 1, the model is sequentially trained until the $t$-th domain is finished, and then we visualize the decision boundary on current domain $\mathcal{D}^t$ and the next future domain $\mathcal{D}^{t+1}$, $t=5,6,7,8$. According to the result, the decision boundary generated by our method successfully adapts to future domains, showing that our method can capture the underlying temporal drift of data. In the revision, we will add this experiment.

---

> ### Author Response · Authors · 2023-08-18
> **Looking forward to seeing your feedback!**
>
> Dear reviewer Nwuj,
>
> We have detailedly discussed **the contribution and memory complexity** of our approach. Additionally, we have provided fresh results encompassing **a recent state-of-the-art CL method**, as well as continuing DA methods (**CIDA and CDOT**), along with **visualizations of evolving decision boundaries**. We hope our responses have addressed your questions and concerns. The rebuttal period is going to end. If you have any other questions or suggestions, please let us know and we are more than happy to respond as soon as possible. Looking forward to your feedback. Thanks so much!
>
> Best regards,
>
> Authors

---

> ### Comment · Reviewer_Nwuj · 2023-08-20
>
> I appreciate the responses from the authors and my previous concerns about the complexity analyses and more empirical results have been addressed. I have increased my score accordingly.

---

### Author Rebuttal · Authors · 2023-08-10

Sincerely thank all reviewers for the efforts in reviewing our paper and the constructive suggestions. We are more than encouraged that reviewers find
* our proposed problem of Continual Domain Generalization over Temporal Drift (CDGTD) to be **interesting** and **novel** (*Reviewer Nwuj*), **innovative** and **meaningful** (*Reviewer P1pd*), **important** (*Reviewer uaS8*),
* our method to be **novel** and **intriguing** (*Reviewer P1pd*), **new** and with **efficacy** (*Reviewer uaS8*), **novel** and **promising** (*Reviewer YTjU*),
* our paper to be **clearly written** (*Reviewer Nwuj*) and with **commendable structure** (*Reviewer YTjU*) and **quality** (*Reviewer P1pd*),
* the ablation study to be **thorough** (*Reviewer P1pd*),
* and the performance to be **state-of-the-art** across a diverse range of tasks (*Reviewer P1pd*).

As for the concerns and suggestions of each reviewer, we have done our utmost to address them and responded to each of them in detail. And a one-page PDF file has been uploaded containing the relevant figures and tables mentioned in the responses. Please refer to the PDF file for detailed results, if needed. Below are some global replies (**R**) that may be used in the  separate response to different reviewers:

> **R1: Introduction to the dataset used in the rebuttal.**

Apart from the dataset in our paper, we additionally run our method on below datasets with temporal shifts that are used by previous temporal DG methods [2, 3]. The description of these datasets are given as below.
* **2-Moons** is a variant of the 2-entangled moons dataset by rotating data counter-clockwise in units of $18^\circ$ to construct 10 domains, where the rotation angle is used to simulate the temporal shift.
* **Online News Popularity (ONP)** summarizes a heterogeneous set of features about articles published by Mashable in a period of two years. The dataset is split into 6 domains by time and the goal is to predict the number of shares in social networks (popularity).
* **Electrical Demand (Elec2)** contains information about the demand of electricity in a particular province. Following [2, 3], the first 30 domains are used (two weeks as one time domain) and the task is to predict if the demand of electricity in each period (of 30 mins) was higher or lower than the average demand over the last day.

For these datasets, following [2, 3], we use the last domain for testing and the rest for training.

> **R2: The complexity anlysis of our method.**

**Memory complexity** mainly comes from the memory pool $\mathcal{M}$. Assuming that there are $T$ historical domains and the feature dimension is $d_f$, then the memory complexity of the memory pool is $\mathcal{O}(T\cdot d_f)$. In practice, the dimension of pooled features is usually much smaller than that of original inputs after processing by the deep neural network. For example, the dimension of an image in the fMoW dataset is $224\times 224 \times 3 = 150,528$, while the dimension of pooled features in DenseNet-121 is $1,024$, about $0.007$ times of the former. Besides, only two vectors need to be stored per domain. Hence, the memory cost of $\mathcal{M}$ is relatively small, compared with sample replay-based CL methods. Concretely, the memory cost of $\mathcal{M}$ for the relatively large dataset fMoW is $32$ MB, and the GPU memory consumption of the whole method EvoS increases by $0.35$ GB over IncFinetune (10.69 GB of IncFinetune and 11.04 GB of EvoS).

**Time complexity** mainly comes from the multi-scale attention module (MSAM). Taking one of the attention module $\mathcal{A}_w$ in MSAM as an example, we assume that $d_f$ is the feature dimension of input tokens, $d_h$ and $n_h$ are the feature dimension and the number of heads, and $n_i$ is the number of input tokens in $\mathcal{A}_w$. Then the time complexity comprises:

* the transformation of input tokens to their query, key and value embeddings: $\mathcal{O}(n_i \cdot d_f \cdot d_h \cdot n_h)$,
* the calculation of attention weight matrix: $\mathcal{O}(n_i \cdot d_h \cdot n_i \cdot n_h)$,
* the multiplication of attention weight matrix and value matrix: $\mathcal{O}(n_i \cdot n_i \cdot d_h \cdot n_h)$,
* convert the feature dimension of attended value embeddings into the input dimension: $\mathcal{O}(n_i \cdot d_h \cdot d_f \cdot n_h)$.

Thus, the time complexity is $\mathcal{O}(n_i \cdot d_f \cdot d_h \cdot n_h) + \mathcal{O}(n_i \cdot d_h \cdot n_i \cdot n_h) + \mathcal{O}(n_i \cdot n_i \cdot d_h \cdot n_h) + \mathcal{O}(n_i \cdot d_h \cdot d_f \cdot n_h)\approx\mathcal{O}((n_i^2 + n_i \cdot d_f) \cdot (\cdot d_h \cdot n_h)).$

Since $n_i$ will be no larger than the number of training domains $T$ and $d_h \cdot n_h$ usually sets to $d_f$ in transformers, the time complexity of MSAM can be roughly approximate as $\mathcal{O}(W \cdot (T^2 d_f + T \cdot d_f^2))$, where $W$ is the number of multi-head attention modules in MSAM. In the implementation, $W$ is set to a relatively small value ($W=3$) in our paper and the time complexity is acceptable.

**Global references:**

[1] Wild-time: A benchmark of in-the-wild distribution shift over time. In NeurIPS, 2022.

[2] Training for the future: A simple gradient interpolation loss to generalize along time. In NeurIPS, 2021.

[3] Temporal domain generalization with drift-aware dynamic neural networks. In ICLR, 2023.

[4] Gobinda Saha, Kaushik Roy. Continual Learning with Scaled Gradient Projection. AAAI, 2023.

---

> ### Comment · Area_Chair_ufVF · 2023-08-18
> **Authors rebuttal**
>
> I acknowledge the authors rebuttal and I am encouraging reviewers to reflect on.

---

### Decision · Program_Chairs · 2023-09-21

**Decision:**

Accept (poster)

**Comment:**

The paper proposes a new practical setting for continual domain generalization where data distribution is evolving over time. The authors have proposed a novel method - multi scale attention. The method is efficient and shows better performance over existing baselines and methods.  The paper is well written and well presented. All reviewers recommend accepting the paper.